# DNA-thioguanine nucleotide as a treatment marker in acute lymphoblastic leukemia patients with *NUDT15* variant genotypes

Hee Young Ju[1], Ji Won Lee[1], Hee Won Cho[1], Ju Kyung Hyun[1], Youngeun Ma[2], Eun Sang Yi[3], Keon Hee Yoo[1], Ki Woong Sung[1], Rihwa Choi[4,5], Hong Hoe Koo[1]*, Soo-Youn Lee[5]*

1 Department of Pediatrics, Samsung Medical Center, Sungkyunkwan University School of Medicine, Seoul, Republic of Korea, 2 Department of Pediatrics, Seoul National University Bundang Hospital, Sungnam, Republic of Korea, 3 Department of Pediatrics, Korea University Guro Hospital, Korea University College of Medicine, Seoul, Republic of Korea, 4 Department of Laboratory Medicine, Green Cross Laboratories, Yongin, Gyeonggi, Republic of Korea, 5 Department of Laboratory Medicine and Genetics, Samsung Medical Center, Sungkyunkwan University School of Medicine, Seoul, Republic of Korea

* hhkoo.koo@samsung.com (HHK); sy117.lee@samsung.com (SYL)

**Data Availability Statement:** All relevant data are within the manuscript and its Supporting information files.

**Funding:** This study was supported by a grant from the Korean Foundation for Cancer Research

## Abstract

### Background

Large inter-individual variations in drug metabolism pose a challenge in determining 6-mercaptopurine (6MP) doses. As the last product of 6MP metabolism, DNA-thioguanine nucleotide (DNA-TGN) could reflect the efficacy of 6MP, especially in patients harboring variants in the 6MP metabolism pathway. The aim of this study was to investigate the clinical significance of DNA-TGN monitoring in Korean pediatric acute lymphoblastic leukemia (ALL) patients, focusing on the *NUDT15* genotype.

### Methods

The subjects of this study were patients who underwent ALL treatment with 6MP. Tests for the *NUDT15* and *TPMT* genotypes were performed, and prospective DNA-TGN and erythrocyte TGN samples were collected after two weeks or more of 6MP treatment. DNA-TGN was quantified using the liquid chromatography-tandem mass spectrometry method.

### Results

A total of 471 DNA-TGN measurements in 71 patients were analyzed, which ranged from 1.0 to 903.1 fmol thioguanine/μg DNA. The 6MP intensity demonstrated a significant relationship with DNA-TGN concentration (*P*<0.001). Patients harboring *NUDT15* variants were treated with a lower dose of 6MP (*P*<0.001); however, there was no significant difference in DNA-TGN concentration when compared to patients carrying wild-type *NUDT15* (*P* = 0.261). These patients also presented higher variation in DNA-TGN levels (*P* = 0.002) and DNA-TGN/6MP intensity (*P* = 0.019) compared to patients carrying wild-type *NUDT15*. DNA-TGN concentration did not show a significant correlation with WBC count (*P* = 0.093).

(KFCR-2017-D-1). The funder provided financial support for the study, but did not have any additional role in the study design, data collection and analysis, decision to publish, or preparation of the manuscript. The specific roles of the authors are articulated in the 'author contributions' section. All authors have no potential conflicts of interest to declare.

**Competing interests:** Author RC is employed by Green Cross Laboratories, but the commercial affiliation does not alter the authors' adherence to PLOS ONE policies on sharing data and materials.

## Conclusions

Patients harboring *NUDT15* variants demonstrated similar DNA-TGN concentrations even at low doses of 6MP and showed high variability in DNA-TGN. Particularly in patients with *NUDT15* variants who need a reduced 6MP dose, DNA-TGN could be applied as a useful marker to monitor the therapeutic effect of 6MP.

## Introduction

Mercaptopurine is one of the key drugs used for maintenance treatment of childhood acute lymphoblastic leukemia (ALL). However, some patients experience toxicity during mercaptopurine treatment, such as severe infections related to bone marrow suppression, in addition to technical issues, such as the need for dose reduction or frequent treatment interruptions. Mercaptopurine is known to have narrow therapeutic indices and it is also known that frequent dose changes or discontinuation of mercaptopurine is associated with increased relapse of ALL [1]. Maintaining an appropriate concentration of thiopurine metabolites is important for preventing relapse or the occurrence of toxic events after ALL treatment.

If accurate metabolite measurements can be used to adjust the dose of 6-mercaptopurine (6MP), ALL patients, especially those with a variety of metabolic variations, can expect better treatment outcomes. To date, the degree of cytopenia and liver function have been used to adjust the dose of 6MP during ALL treatment. However, the metabolism of 6MP varies greatly, and the dosage may vary by up to 10 fold between patients; this uncertainty makes it difficult to successfully carry out therapeutic adjustments using these classical methods [2]. Moreover, it is known that varying 6MP doses and 6MP drug interruptions are related to varying thioguanine nucleotide (TGN) levels and an increase in relapse rate [1]. Other studies have evaluated the use of erythrocyte TGN (RBC TGN) as an indicator of 6MP metabolism, but these studies have found that RBC TGN is not a robust prognostic marker for relapse [3]. Recently, DNA-incorporated thioguanine nucleotides (DNA-TGNs) have been proposed as the primary mediators of 6-mercaptopurine cytotoxicity [4]. In addition, a large-scale prospective study by the Nordic Society of Pediatric Hematology and Oncology (NOPHO) group demonstrated that the concentration of DNA-TGN is negatively correlated with relapse [5].

Several genes have been linked to the metabolism of 6MP used in the treatment of ALL, including *TPMT* and *NUDT15* [6]. Variants of these two genes have been linked to an increased risk of leukopenia during 6MP treatment [7, 8]. Patients carrying variant *TPMT* or *NUDT15* alleles, which exhibit decreased activity compared to the wild type, experience increased toxicity even at low doses of 6MP. Particularly, *NUDT15* variants have been shown to be more prevalent in Asian populations than in other ethnic groups [9–11]. As a nucleotide diphosphatase, NUDT15 is known to inactivate thiopurine metabolites by converting TGTP to TGMP and therefore negatively regulates the cytotoxic effects of this class of drugs [12]. It can then be assumed that in individuals harboring *NUDT15* variants, even low doses of 6MP would produce relatively high DNA-TGN concentrations, thereby allowing for effective treatment.

To date, three papers have been published on DNA-TGN levels, covering patients with *NUDT15* variant genotypes [13–15]. These studies have reported that patients with the *NUDT15* variant genotype show higher DNA-TGN levels and toxicity when 6MP is administered at regular doses. In contrast, the present study focused on DNA-TGN in patients with *NUDT15* or *TPMT* variant genotypes who were treated with reduced doses of 6MP. In addition, in this study, since repeated blood tests were performed in patients continuously receiving 6MP, changes in DNA-TGN over time could be observed.

The aim of this study was to investigate the pattern of DNA-TGN when 6MP is administered at reduced doses in patients harboring *NUDT15* variants (dynamically adjusted based on the degree of cytopenia). It was designed to evaluate whether DNA-TGN concentration differs based on the genotype of *NUDT15*, and whether DNA-TGN levels reflect treatment-related toxicity during ALL treatment. In addition, the variation of DNA-TGN during maintenance treatment with 6MP was explored.

## Methods

### 1. Study patients

In this study, the participants were recruited from patients undergoing 6MP-based treatment for ALL or lymphoma at the Department of Pediatrics, Samsung Medical Center, Seoul, Republic of Korea. The enrollment period was from January 2018 to February 2020, and patients with newly diagnosed as well as relapsed ALL or lymphoma were included in the study. Patients who were not expected to receive 6MP treatment and those who did not wish to participate in the study were excluded. The number of participants in the study was not set in advance because of the exploratory study design of DNA-TGN in patients with an existing treatment schedule.

The present study protocol was reviewed and approved by the Institutional Review Board of Samsung Medical Center (approval No. 2017-11-161). Informed consent was obtained from all subjects when they were enrolled. Written consent was obtained from the parents of minors and from patients for adults.

### 2. Treatment

Patients were treated in accordance with the Children's Oncology Group (COG) regimen-based treatment. Regardless of their risk group, patients who received 6MP for two or more weeks were enrolled in this study. The initial dose for 6MP maintenance therapy was determined based on the lowest 6MP dose that did not result in cytopenia (described in previous treatment cycles, including consolidation, interim maintenance, or delayed intensification). The 6MP dosage was rounded up to the nearest available dose for administration, and the dosage was gradually increased upon identifying that 6MP was tolerable. The intended final dose of 6MP for patients harboring wild-type *NUDT15* and *TPMT* was set at 50 mg/m$^2$/day. 6MP intensity was defined as the ratio between the prescribed daily 6MP dose (mg/m$^2$/day) and the standard daily dose (50 mg/m$^2$/day).

### 3. Sample collection and analysis

Prospective metabolite samples (DNA-TGN, RBC TGN, methylmercaptopurine nucleotides (MMPN)) were acquired simultaneously after continuous administration of 6MP for at least 14 days after initiation of constant-dosing 6MP treatment: after 2 weeks for the protocol in which patients took 6MP for 2 weeks; after 2 and 4 weeks for the protocol in which patients took 6MP for 4 weeks; after 2, 4, and 8 weeks for the protocol in which patients took 6MP for 8 weeks; and after 2 and 4 weeks of the 1$^{st}$ cycle of maintenance treatment. Blood sampling for DNA-TGN was performed along with routine sampling to monitor complete blood cell count and chemistry tests during chemotherapy. If the 6MP dose was changed or if the treatment was stopped, additional samples were collected. DNA- and RBC- TGN concentrations and MMPN were measured using liquid chromatography-tandem mass spectrometry (LC-MS/MS). The DNA-TGN test was performed with isotope-labeled TG-d3 and guanine-d3 as internal standards, as previously described. Chromatographic etheno-TG peaks were normalized

using etheno-G by calculating TG responses as etheno-TG area/etheno-G area (DNA-TG = [etheno-TG response/etheno-G response]/[etheno-TG-d3 response/etheno-G-d3 response]) [16, 17]. In this study, blood samples were collected prospectively; however, the DNA-TGN test was not performed in real time. Therefore, the dose of 6MP was adjusted according to white blood cell (WBC) count, aspartate aminotransferase (AST), alanine transaminase (ALT), but not DNA-TGN values. Genotyping of *NUDT15* and *TPMT* was performed before the start of the 6MP maintenance therapy via direct sequencing. Direct Sanger sequencing of exons 1 and 3 of *NUDT15* and direct sequencing of exons 3 to 10 of *TPMT* were performed after PCR [18, 19]. Since *NUDT15* *1/*2 and *3/*6 genotypes were not distinguished technically by this method, *1/*2 represents *1/*2 or *3/*6 [18].

## 4. Data collection

Clinical data were collected upon diagnosis and included 6MP dose, treatment interruption, relapse, survival, and treatment-related toxicity. Laboratory results were collected for WBC, absolute neutrophil count (ANC), platelet count, AST, ALT, and total bilirubin.

## 5. Statistical analysis

Descriptive analyses were conducted for age, sex, diagnosis, *NUDT15*, *TPMT* genotype, 6MP dose, DNA-TGN, and RBC TGN. Fisher's exact test was performed to determine differences in sex, diagnosis, and relapse status according to genotypes. Correlation analysis was performed to determine the correlation of DNA-TGN by 6MP dose and WBC count by DNA-TGN. Fisher's exact test was used to determine the differences in the rate of toxicity in the *NUDT15* variant and wild type groups. For the association of DNA-TGN by 6MP dose and hematologic toxicity, analyses were conducted only on the samples collected during maintenance treatment to rule out the effects of other drugs. The Kruskal-Wallis test was performed to compare the differences in DNA-TGN according to sex, diagnosis, and relapse state. In addition, the same test was used to compare the differences in DNA-TGN, 6MP intensity, and DNA-TGN/6MP intensity according to genotypes. To determine the difference in DNA-TGN/6MP intensity according to genotype, normalization of DNA-TGN/6MP intensity was done. To evaluate the distribution of DNA-TGN, DNA-TGN/6MP intensity and RBC TGN values between and within patients, coefficients of variation (CV) were calculated. Linear mixed model analysis was performed to compare the pattern of change between *NUDT15* variant and wild type groups. When performing CV and linear mixed model analyses to determine the degree of variability, only the values collected during the maintenance treatment of patients from which were taken DNA-TGN samples more than three times during the maintenance period were used.

All statistical analyses were performed using SPSS v25 (SPSS Inc., Chicago, IL, USA), and all graphical representations were prepared using Prism 7 (Graph Pad Software, San Diego, CA, USA). All statistical tests used two-sided probability, and significance was set at $P < 0.05$.

## Results

### 1. Patients and samples

A total of 72 patients were enrolled in this study, but one patient was excluded from the analysis due to a lack of clinical information. Of the 71 patients in this study, 47 were male and 24 were female; 19 patients harbored an *NUDT15* variant and 3 patients harbored a *TPMT* variant. None of the patients harbored both genetic variants. Detailed clinical information is presented in Table 1.

**Table 1. Participant and DNA-TGN sample characteristics.**

| Characteristic | Total Patients[*] (N = 71) | *NUDT15* WT[†], *TPMT* WT (N = 49) | *NUDT15* variant, *TPMT* WT (N = 19) | *NUDT15* WT, *TPMT* variant (N = 3) | *P* value |
|---|---|---|---|---|---|
| Sex | | | | | |
| Male | 47 | 35 | 11 | 1 | 0.235 |
| Female | 24 | 14 | 8 | 2 | |
| Median age | 6.9 | 6.5 | 8.5 | 8.9 | 0.441 |
| (range) | (1.2–23.3) | (1.2–21.2) | (2.4–23.3) | (4.9–11.3) | |
| Diagnosis | | | | | |
| B-ALL[‡] | 59 | 41 | 15 | 3 | 0.522 |
| T-ALL | 7 | 5 | 2 | 0 | |
| B-NHL[§] | 1 | 1 | 0 | 0 | |
| T-NHL | 4 | 2 | 2 | 0 | |
| Previous relapse | | | | | |
| Yes | 68 | 48 | 18 | 2 | 0.078 |
| No | 3 | 1 | 1 | 1 | |
| Variant genotypes | | | *NUDT15* <br> *1/*2, n = 8 <br> *1/*3, n = 4 <br> *1/*5, n = 3 <br> *1/*6, n = 3 <br> *2/*3, n = 1 | *TPMT* <br> *1/*3, n = 3 | N/A |
| On-therapy | 48 | 35 | 11 | 2 | 0.564 |
| Off-therapy | 23 | 14 | 8 | 1 | |
| Relapse after enrolment | 1 | 1 | 0 | 0 | N/A |
| Death | 0 | 0 | 0 | 0 | N/A |
| Number of DNA-TGN samples taken | 471 | 291 | 167 | 13 | 0.858 |
| Mean number of DNA-TGN samples for each patient | 6.6 | 5.9 | 8.8 | 4.3 | N/A |
| | (range, 1–21) | (range, 1–16) | (range, 1–21) | (range, 1–10) | |
| Timepoint at which DNA-TGN sample was collected | | | | | |
| Consolidation | 20 | 16 | 4 | 0 | 0.153 |
| Interim maintenance | 86 | 60 | 25 | 1 | |
| Delayed intensification | 6 | 2 | 4 | 0 | |
| Maintenance | 359 | 213 | 134 | 12 | |
| Median 6MP intensity during maintenance treatment | 0.33 (range, 0.01–2.3) | 0.4 (range, 0.07–2.3) | 0.22 (range, 0.02–1.43) | 0.03 (range, 0.01–0.52) | <0.001 |

[*]There was no patient harboring both *NUDT15* and *TPMT* variants;

[†]WT, wild type;

[‡]ALL, acute lymphoblastic leukemia;

[§]NHL, non-Hodgkin lymphoma.

A total of 472 DNA-TGN samples were collected. As one sample was collected during the treatment interruption period, only 471 samples were used for analysis. The mean number of DNA-TGN samples collected per patient was 6.6 (range, 1–21). A total of 359 measurements were performed during the maintenance therapy, with 112 measurements taken before the initiation of the maintenance therapy. DNA-TGN levels from whole measurements ranged from 1.0 to 903.1 fmol TG/μg DNA (fmol thioguanine/μg DNA), with a median level of 119.0 fmol TG/μg DNA.

The median follow-up time was 18.1 months (range, 3.2–27.3 months). At the time of the last data collection, 23 patients were off therapy, and 48 patients were on therapy. There were no deaths during this study, but one patient exhibited a relapse of ALL. This patient was sent for DNA-TGN evaluation during the consolidation treatment, and relapse was confirmed immediately after that. The treatment protocol was changed, and hematopoietic stem cell transplantation was performed. Relapse was not observed in patients who underwent ALL maintenance treatment.

When comparing the distribution of DNA-TGN, male (vs. female, $P = 0.001$) and Pre-B ALL (vs. Pre-T ALL, lymphoma, $P<0.001$) showed higher levels of DNA-TGN. However, when comparing DNA-TGN/6MP intensity, there was no significant difference according to sex ($P = 0.18$), disease ($P = 0.061$), or previous relapse ($P = 0.675$).

## 2. DNA-TGN results stratified based on the *NUDT15* status

The distribution of DNA-TGN levels was stratified based on the genotype of *NUDT15* and *TPMT* (Fig 1A). There was a significant positive correlation between DNA-TGN levels and 6MP dosage in all patients ($P<0.001$). When examining the distribution of DNA-TGN according to the 6MP dose by the subgroup of *NUDT15* variants, *NUDT15* *1/*2, *1/*3, and *1/*5, showed higher DNA-TGN/6MP intensity than that of wild-type patients. However, there was no statistically significant difference between *NUDT15* *1/*6 or *TPMT* variant genotype patients and wild-type patients, but the level was lower in one DNA-TGN test performed on the *NUDT15* *2/*3 genotype patient (Fig 1B).

Patients with *NUDT15* variant genotypes were treated with significantly lower 6MP intensities when compared to patients with the wild-type *NUDT15* genotype ($P<0.001$). (Fig 2A) However, there was no significant difference in the DNA-TGN levels between patients with *NUDT15* variant genotypes, *TPMT* variant genotypes, and both wild types ($P = 0.261$) (Fig 2B). The ratio of DNA-TGN/6MP intensity was significantly higher in patients harboring *NUDT15* variants than that in patients harboring the wild-type *NUDT15* ($P<0.001$); however, no difference was observed between patients harboring *TPMT* variants and the wild type ($P = 0.323$) (Fig 2C).

The median DNA-TGN level during the maintenance period in patients harboring both *NUDT15* and *TMPT* wild-type genes was 126.2 fmol TG/μg DNA. In patients harboring *NUDT15* variants, 73.8% of the 6MP dose was needed to reach the median DNA-TGN level compared to patients harboring wild-type *NUDT15*.

## 3. Toxicity during treatment

Of the 63 patients who underwent maintenance therapy, eight patients manifested leukopenia (WBC < 1,500/μL) at the time of DNA-TGN sampling. Among these patients, four harbored *NUDT15* variants (two *1/*2, one *1/*5, one *1/*6 genotype), and no patient with a *TPMT* variant experienced cytopenia. The DNA-TGN concentrations during these leukopenia episodes ranged from 27.8 to 504.8 fmol TG/μg DNA. Three of the patients developed leukopenia before 12 weeks of treatment, and their 6MP intensities were 0.17, 0.12, and 0.08, respectively. However, in the five cases of leukopenia that occurred after 12 weeks of 6MP-based maintenance therapy, the 6MP intensity ranged from 0.22 to 0.98 (median 0.4).

Three DNA-TGN samples were collected when treatment was discontinued for reasons other than cytopenia (AST or ALT elevation in two cases, fever with viral infection in one case). More detailed information on the measurements associated with cytopenia or toxicities resulting from treatment interruptions are described in Table 2.

(A)

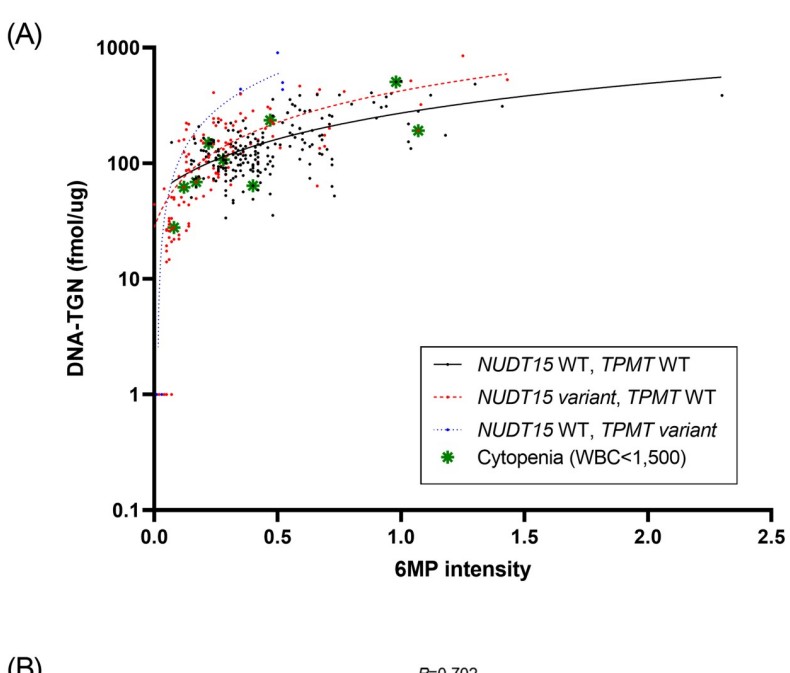

(B)

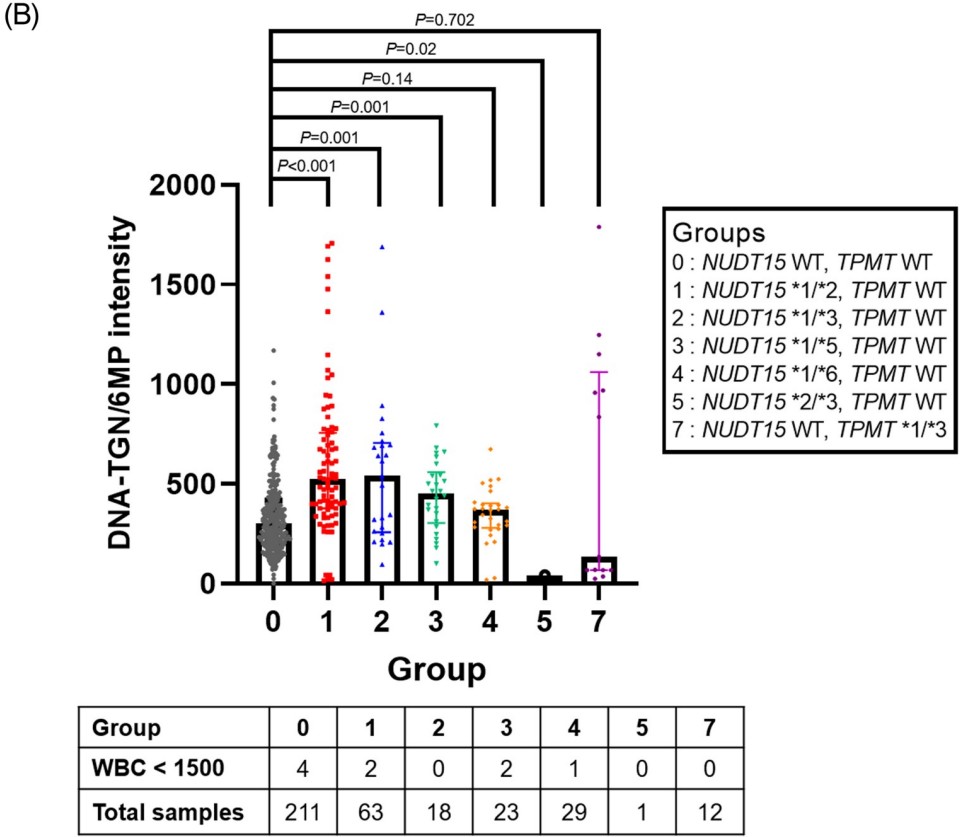

| Group | 0 | 1 | 2 | 3 | 4 | 5 | 7 |
|---|---|---|---|---|---|---|---|
| WBC < 1500 | 4 | 2 | 0 | 2 | 1 | 0 | 0 |
| Total samples | 211 | 63 | 18 | 23 | 29 | 1 | 12 |

**Fig 1. DNA-TGN concentration versus 6MP intensity.** (A) stratified based on the genotype of *NUDT15* and *TPMT* (y-axis, DNA-TGN in log scale; WT, wild type). (B) DNA-TGN/6MP intensity ratio and rate of cytopenia according to pharmacogenetic subgroups. Line represents median with interquartile range.

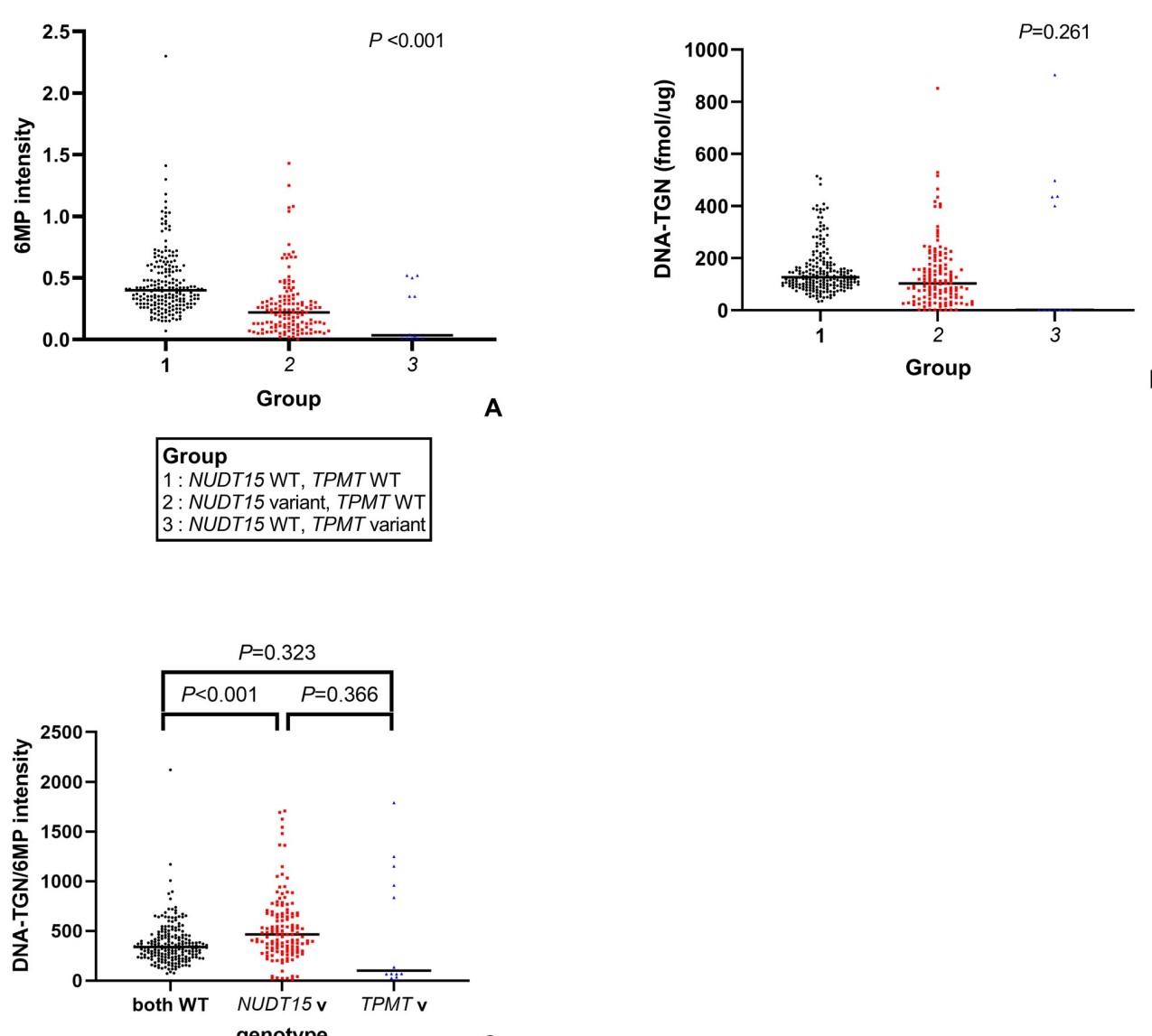

**Fig 2. Distribution of various parameters based on genotype.** (A) 6MP intensity (B) DNA-TGN (C) DNA-TGN/6MP intensity (WT, wild type; v, variant).

When these patients were excluded, the WBC counts for all the other patients were within the reference range. Within the total 359 measurements gathered during maintenance treatment, WBC count did not exhibit a significant correlation with the DNA-TGN level ($P = 0.093$) or RBC TGN ($P = 0.425$). When analyzing separately the patients harboring *NUDT15* variants, the DNA-TGN level also showed no association with WBC ($P = 0.87$). Among the 17 measurements with the highest DNA-TGN scores of 400 or more, cytopenia occurred in only one case.

## 4. Inter-individual variability

To evaluate the degree of change in DNA-TGN within each patient, the CV for DNA-TGN was calculated for each patient and its distribution was examined. CV was checked in 51

**Table 2. Cases with toxicity related to treatment.**

| Patient | *NUDT15* | *TPMT* | Time of Sample | 6MP intensity* | MTX intensity† | DNA-TGN (fmol TG/μg DNA) | RBC TGN (μmol/L) | MMPN‡ (μmol/L) | WBC (/μL) | ANC (/μL) | Reason of interruption |
|---|---|---|---|---|---|---|---|---|---|---|---|
| *Cytopenia* | | | | | | | | | | | |
| 1 | *1/*1 | *1/*1 | 10th cycle | 0.98 | 1.01 | 504.8 | 364.7 | 35039.8 | 1280 | 540 | - |
| 2 | *1/*1 | *1/*1 | 2nd cycle | 0.40 | 0.97 | 63.8 | 200 | 1150.3 | 1260 | 200 | - |
| 3 | *1/*1 | *1/*1 | 3rd cycle | 0.22 | 1.03 | 149.1 | 385.7 | 469.3 | 780 | 390 | - |
| 4 | *1/*1 | *1/*1 | 3rd cycle | 0.28 | 0.89 | 107.6 | 261.9 | 610.5 | 1440 | 310 | - |
| 5 | *1/*5 | *1/*1 | 7th cycle | 0.47 | 1.04 | 235.9 | 313.2 | 1057.1 | 1440 | 910 | - |
| 6 | *1/*2 | *1/*1 | 1st cycle (2 weeks from start) | 0.17 | 0.69 | 68.7 | 65.9 | 1080.5 | 1410 | 940 | - |
| 7 | *1/*2 | *1/*1 | 1st cycle (6 weeks from start) | 0.12 | 0.61 | 62.0 | 70.4 | 154.6 | 770 | 270 | - |
| 8 | *1/*6 | *1/*1 | 1st cycle (10 weeks from start) | 0.08 | 0.98 | 27.8 | 51.5 | 85.4 | 1040 | 320 | - |
| *Treatment interruption for reasons other than cytopenia* | | | | | | | | | | | |
| 9 | *1/*1 | *1/*1 | 3rd cycle | 0.28 | 0.94 | 90.0 | 236.3 | 791.9 | 2420 | 1310 | AST§ 1040 |
| 10 | *1/*1 | *1/*1 | 2nd cycle | 0.47 | 0.82 | 80.5 | 208.9 | 2064.9 | 2400 | 830 | AST§ 1145 |
| 11 | *1/*2 | *1/*1 | 4th cycle | 0.18 | 0.97 | 74.8 | 94.5 | 413.2 | 5210 | 4120 | Fever with viral infection |

*6MP intensity, 6MP dose (mg/m$^2$/day) / 50 mg/m$^2$/day;

†MTX intensity, Methotrexate dose (mg/m$^2$/week) / 20 (mg/m$^2$/week);

‡MMPN, methylmercaptopurine nucleotides;

§AST, aspartate aminotransferase.

patients who provided at least three or more DNA-TGN samples during the maintenance period. Among the three patients with *TPMT* variant genotypes, only one was tested for DNA-TGN multiple times; therefore, patients with *TPMT* variant genotypes were not included in the CV analysis. The CV for DNA-TGN ranged from 13.9% to 76.1% in patients harboring wild-type *NUDT15*, with a median value of 32.7%. In patients carrying *NUDT15* variants, the CV ranged from 29.5% to 104% and the median value was 51.4%, which were significantly higher than those in patients harboring wild-type *NUDT15* (*P* = 0.001). The CV for RBC TGN ranged from 29.5% to 143.6% (median 53.5%) in patients harboring *NUDT15* variants, and 13.7% to 76.1% in patients harboring wild-type *NUDT15* (median 36.2%) (*P* = 0.009). As this large variation in DNA-TGN and RBC TGN may have been due to frequent changes in the 6MP dose in the *NUDT15* variant group, evaluation of the CV in DNA-TGN/6MP intensity was also performed to eliminate the effect of dose variation. The CV for the DNA-TGN/6MP intensity ratios ranged from 23.8% to 90.5% (median 45.4%) in patients harboring the *NUDT15* variants, which was significantly higher (*P* = 0.019) than that in patients harboring wild-type *NUDT15* (5.6% to 71.8%, median 32.3%).

## 5. Changes in 6MP dose during the maintenance treatment

Changes in the 6MP dose were analyzed in patients who underwent maintenance therapy for more than three months. In total, 34 patients with both wild-type *NUDT15* and *TPMT*, 11 patients harboring *NUDT15* variants, and two patients harboring *TPMT* variants had

undergone maintenance therapy for 3 months or longer. The median starting doses of 6MP were 21.43 mg/m$^2$, 9.83 mg/m$^2$ and 0.89 mg/m$^2$, respectively ($P<0.001$).

The last 6MP dose was compared between the groups based on the assumption that the last 6MP dose was the highest tolerable dose. There were significant differences in the last doses of 6MP between the 3 groups (median dose of 28.8 vs. 13.5 vs. 17.7 mg/m$^2$ in the wild type, *NUDT15* variant, and *TPMT* variant groups, respectively; $P = 0.037$) (Fig 3A). Five patients took a higher dose of 6MP than the intended dose at the end of the treatment due to a high WBC count.

The change in 6MP intensity during maintenance therapy was plotted based on *NUDT15* status (Fig 3B). In addition, the change in DNA-TGN/6MP intensity during the maintenance therapy was plotted against *NUDT15* status (Fig 3C), showing high variability in patients harboring the *NUDT15* variants. However, the linear mixed model analysis did not reveal any significant differences in the pattern of DNA-TGN concentration with respect to time, between patients harboring wild-type *NUDT15* and variants ($P = 0.238$).

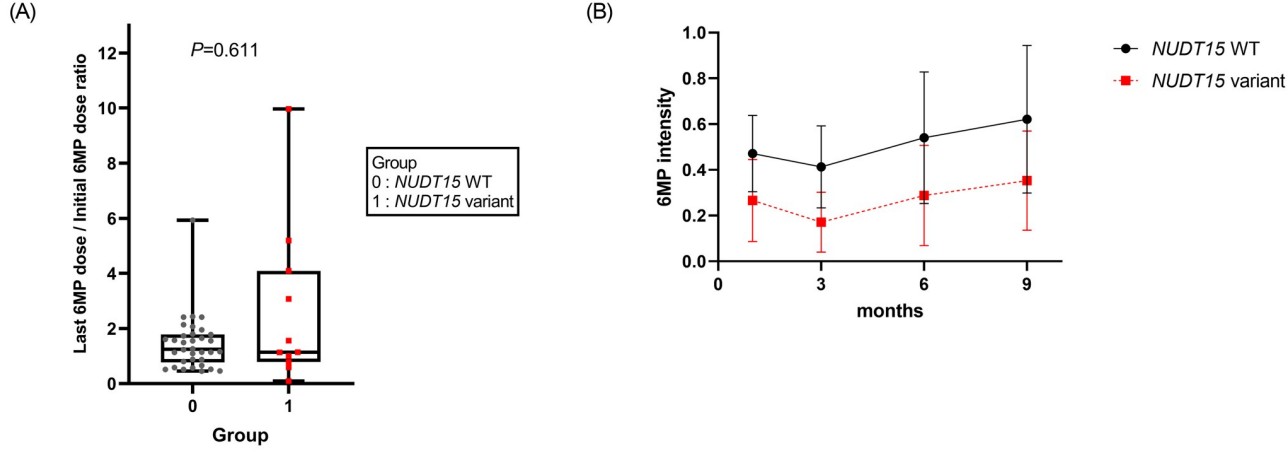

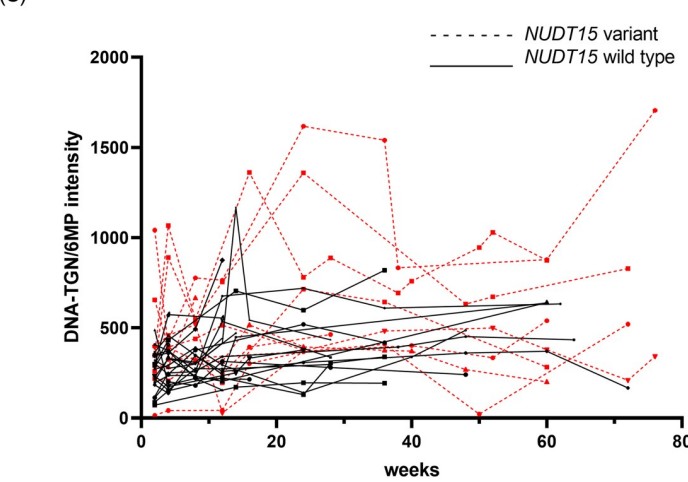

**Fig 3. Changes in 6MP dosage according to *NUDT15* status.** (A) Ratio of starting 6MP dose and last 6MP dose of maintenance treatment. (B) Dose changes in patients harboring wild-type *NUDT15* and *NUDT15* variants. (C) Changes in the DNA-TGN/6MP intensity ratios over the course of the maintenance treatment.

## Discussion

As the last product of 6MP metabolism, DNA-TGN could be used as a marker for evaluating the efficacy of 6MP, especially in patients harboring pharmacogenetic variants of genes coding for proteins involved in the 6MP metabolism pathway. In patients harboring *NUDT15* variants, DNA-TGN accumulates, leading to treatment-related toxicity. Following the publication of the first report linking *NUDT15* polymorphisms to thiopurine-induced leukopenia in a Korean study on patients with inflammatory bowel disease [7], a high incidence of *NUDT15* polymorphism has been reported in East Asian populations. In addition, treatment-related toxicity was reported to be high in ALL patients harboring *NUDT15* polymorphisms [10, 20, 21].

A preclinical study using *NUDT15* knockout mice showed that a reduction in the dose of 6MP resulted in decreased treatment-related toxicity without attenuating treatment outcomes. In the study, DNA-TGN levels in *NUDT15* knockout mice (which corresponds to *NUDT15* homozygous variant) were similar to those in *NUDT15* wild-type mice treated with a normal dose of 6MP [14]. However, tolerable doses of 6MP have been shown to vary considerably in patients with heterozygous variant for *NUDT15* as compared to those with homozygous variant for *NUDT15*. The Clinical Pharmacogenetics Implementation Consortium (CPIC) guidelines suggest initiating 6MP treatment at a reduced dose in patients with a *NUDT15* variant genotype, and then adjusting the 6MP dose based on the observed degree of myelosuppression [22]. However, while this dosing method can prevent toxicity, it is difficult to predict the treatment effects.

In this study, DNA-TGN samples were serially collected from patients with ALL receiving 6MP treatment in a prospective manner and were then subjected to a variety of in-depth analyses.

First, the correlation between DNA-TGN and 6MP intensity was examined based on *NUDT15* status. Patients harboring *NUDT15* variants exhibited higher DNA-TGN/6MP intensity ratios, consistent with the findings of a previous report [13]. Three previously published papers have demonstrated that higher DNA-TGN concentrations coincided with a higher incidence of cytopenia in patients harboring *NUDT15* variants when the patients received the standard dose of 6MP [13–15]. In contrast, in this study, patients harboring the *NUDT15* variants began their treatment of 6MP at a reduced dose, and it was estimated that an equivalent median DNA-TGN concentration could be achieved using 73.8% of the 6MP dose used in patients harboring wild-type *NUDT15*. This result suggests that in patients harboring *NUDT15* variants, thiopurine dose-reduction strategies reduce toxicity while not reducing the therapeutic effect.

Second, the distribution and changing patterns in DNA-TGN concentrations were evaluated based on the genotypes of *NUDT15*. The results demonstrate that DNA-TGN and RBC TGN tend to be more variable in patients with the *NUDT15* variant genotype. As frequent 6MP dose changes in patients with the *NUDT15* variant genotype could cause some changes in the DNA-TGN and RBC TGN values, variation in DNA-TGN/6MP intensity was also evaluated. DNA-TGN/6MP intensity ratios were also more variable in patients harboring *NUDT15* variants than those in patients harboring wild-type *NUDT15*. A previous COG study reported a high risk of relapse in ALL patients with large RBC TGN variability [1]. In that study, variability in drug dosing and treatment interruptions (possibly to maintain ANC within a certain range) were thought to be associated with varying levels of TGN. Patients harboring *NUDT15* variants who begin treatment of 6MP at a reduced dosage and are then gradually acclimatized to increasing doses over time, also show large variations in DNA-TGN and RBC TGN levels. This suggests that while adjusting 6MP dose during treatment, frequent measurement of metabolites such as DNA-TGN should be performed to improve overall therapeutic efficacy.

Thiopurine is known to have a delayed cytotoxic effect. At least two or more S phases of cell cycles are needed for 6TGN to incorporate into DNA, and this 6TGN-substituted DNA replicates and presents mismatches, eventually leading to cell death [4]. Considering the timing of the toxicity onset, it is helpful to test DNA-TGN after 2 weeks or more, when the dose of 6MP is changed, or when toxicity occurs, rather than immediately after starting 6MP.

Third, the DNA-TGN levels in relation to the clinical observations were evaluated. The results of this study did not show any correlation between DNA-TGN or RBC TGN and WBC counts. This may be because the treatment was interrupted in only a small number of recruited patients due to the development of cytopenia. In this study, the DNA-TGN test was performed at the beginning of the maintenance period in many cases. During this period, 6MP was administered at a low dose, and therefore, only a few cases presented cytopenia. Other factors that may cause hematologic toxicity—simultaneously administered medications and infection—should also be considered in the interpretation of results. Since methotrexate (MTX) is continuously administered in addition to 6MP during maintenance treatment, treatment-related toxicity can be caused by both drugs. However, in this study, MTX was not included in the analysis because it was considered that there would be little variation in toxicity due to MTX, as the dose of MTX was kept near constant in each patient. Previous studies have reported that there is a difference in 6MP tolerance according to sex [23], but this study did not show a difference in DNA-TGN values compared to 6MP intensity according to sex.

The appropriate target therapeutic range of DNA-TGN concentration is not yet known. To determine the target therapeutic range of DNA-TGN, long-term follow-up with variable 6MP doses would be needed to determine the relationship between DNA-TGN and clinical outcomes, including relapse and toxicity. In addition to the few studies already performed regarding DNA-TGN, this study can help establish the standard for therapeutic dose adjustment of 6MP based on DNA-TGN.

## Study strength and limitations

This appears to be the first study to analyze prospective serial DNA-TGN samples in a large number of patients—including those, which carry *NUDT15* variant genotypes—treated with reduced doses of 6MP. In addition, this is the first study to report detailed clinical data for these patients. However, this study has several limitations. The overall rate of cytopenia was low due to the strategy of starting 6MP treatment at a low dose and increasing the dose in patients that tolerated it well. Therefore, it was difficult to estimate the exact concentration of DNA-TGN at which toxicity occurs. In addition, as the follow-up period was short, it was impossible to evaluate the relationship of clinical outcomes including relapse, survival, and long-term complications, with DNA-TGN level. Since drug metabolism may differ in relapsed patients, it would be better to investigate relapsed patients separately if the number of patients is sufficient [24].

## Conclusion

In conclusion, this study demonstrated that DNA-TGN positively correlates with the dose of 6MP, and that DNA-TGN concentration can be maintained at levels similar to those observed in patients carrying wild-type *NUDT15*—even at lower doses of 6MP—in patients harboring *NUDT15* variants. In addition, DNA-TGN concentrations showed high variability in patients harboring *NUDT15* variants, which suggests the need for close DNA-TGN monitoring to provide more finely tuned treatment. This study was able to demonstrate the efficacy of DNA-TGN concentration in monitoring the treatment effects of 6MP, particularly in patients carrying *NUDT15* variant genotypes who need a reduced 6MP dose. Further prospective

studies are needed to determine whether treatment outcomes can be improved by employing DNA-TGN values to titrate 6MP doses in patients harboring wild-type and variant *NUDT15*.

## Supporting information

**S1 Dataset.**
(XLSX)

**S1 Appendix. Detailed description of study methods.**
(DOCX)

## Author Contributions

**Conceptualization:** Ji Won Lee, Hong Hoe Koo, Soo-Youn Lee.

**Data curation:** Hee Young Ju, Hee Won Cho, Ju Kyung Hyun, Youngeun Ma, Eun Sang Yi, Keon Hee Yoo, Ki Woong Sung, Hong Hoe Koo.

**Funding acquisition:** Ji Won Lee, Soo-Youn Lee.

**Investigation:** Hee Young Ju, Ji Won Lee, Rihwa Choi.

**Methodology:** Rihwa Choi, Soo-Youn Lee.

**Software:** Hee Young Ju.

**Supervision:** Hong Hoe Koo, Soo-Youn Lee.

**Validation:** Rihwa Choi.

**Writing – original draft:** Hee Young Ju.

**Writing – review & editing:** Hee Young Ju, Ji Won Lee, Hee Won Cho, Ju Kyung Hyun, Youngeun Ma, Eun Sang Yi, Keon Hee Yoo, Ki Woong Sung, Rihwa Choi, Hong Hoe Koo, Soo-Youn Lee.

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
