## [Decision Letter · Decision Letter 0]

20 Oct 2020

PONE-D-20-27853

DNA-thioguanine nucleotide as a treatment marker in acute lymphoblastic leukaemia patients with NUDT15 variant genotypes

PLOS ONE

Dear Dr. Koo,

Thank you for submitting your manuscript to PLOS ONE. After careful consideration, we feel that it has merit but does not fully meet PLOS ONE’s publication criteria as it currently stands. Therefore, we invite you to submit a revised version of the manuscript that addresses the points raised during the review process.

As appended below, the reviewers have raised major concern/critique (Reviewer # 1 is against publication) and suggested further justification/work to consolidate the findings. Do go through the comments and amend the MS accordingly. What is the novel aspect of the present study? This MUST be stated clearly in the text. After incorporating all the amendments the MS should be checked by native speaker for grammar and syntax errors.

We look forward to receiving your revised manuscript.

Kind regards,

A. M. Abd El-Aty

Academic Editor

PLOS ONE

3. You indicated that you had ethical approval and informed consent for your study.

In your Methods section, please ensure you have also stated whether you obtained consent from parents or guardians of the minors included in the study or whether the research ethics committee or IRB specifically waived the need for their consent.

4.  Please provide a sample size and power calculation in the Methods, or discuss the reasons for not performing one before study initiation.

5. In your Methods section, please provide additional information about the participant recruitment method and the demographic details of your participants.

Please ensure you have provided sufficient details to replicate the analyses such as:

a) a description of any inclusion/exclusion criteria that were applied to participant recruitment,

b) the methods used to collect patient samples.

6. Please ensure your methods are described in sufficient detail for others to replicate the study.

Specifically, please provide a brief summary of the methods involved in LC-MS and genotyping.

7. PLOS requires an ORCID iD for the corresponding author in Editorial Manager on papers submitted after December 6th, 2016. Please ensure that you have an ORCID iD and that it is validated in Editorial Manager. To do this, go to ‘Update my Information’ (in the upper left-hand corner of the main menu), and click on the Fetch/Validate link next to the ORCID field. This will take you to the ORCID site and allow you to create a new iD or authenticate a pre-existing iD in Editorial Manager. Please see the following video for instructions on linking an ORCID iD to your Editorial Manager account: https://www.youtube.com/watch?v=_xcclfuvtxQ

8. Thank you for stating the following in the Financial Disclosure section:

'This study was supported by a grant from the Korean Foundation for Cancer Research (KFCR-2017-D-1). The funders had no role in study design, data collection and analysis, decision to publish, or preparation of the manuscript.'

We note that one or more of the authors are employed by a commercial company: Green Cross Laboratories

Editor's comments:

As appended below, the reviewers have raised major concern/critique (Reviewer # 1 is against publication) and suggested further justification/work to consolidate the findings. Do go through the comments and amend the MS accordingly. What is the novel aspect of the present study? This MUST be stated clearly in the text. After incorporating all the amendments the MS should be checked by native speaker for grammar and syntax errors.

Reviewers' comments:

Reviewer's Responses to Questions

**Comments to the Author**

1. Is the manuscript technically sound, and do the data support the conclusions?

Reviewer #1: Partly

Reviewer #2: Partly

Reviewer #3: Yes

Reviewer #4: Yes

2. Has the statistical analysis been performed appropriately and rigorously? 

Reviewer #1: Yes

Reviewer #2: No

Reviewer #3: No

Reviewer #4: Yes

3. Have the authors made all data underlying the findings in their manuscript fully available?

Reviewer #1: No

Reviewer #2: No

Reviewer #3: Yes

Reviewer #4: Yes

4. Is the manuscript presented in an intelligible fashion and written in standard English?

Reviewer #1: Yes

Reviewer #2: Yes

Reviewer #3: Yes

Reviewer #4: Yes

5. Review Comments to the Author

Reviewer #1: The article investigated the amount of thioguanine incorporated into DNA (DNA-TGN) of mononuclear cells in patients receiving maintenance chemotherapy for acute lymphoblastic leukemia, and emphasized the importance of DNA-TGN monitoring in assessing the toxicity and efficacy of 6-mercaptopurine among those harboring NUDT15 variants.

The large parts of results are already reported in the previous reports, but they found high variability in DNA-TGN in patients with NUDT15 variant genotypes, which possibly lead to negative impacts on the outcome of leukemia.

The author concluded that DNA-TGN monitoring may be superior to the combination of the genomic diagnosis of NUDT15 genomic variants and 6MP dosing based on the results of genomic analysis for NUDT15 and TPMT.

The conclusion is not completely proved with the results of this study.

For example, is the amount of DNA-TGN at one point sometime after 6MP-administration enough to assess the 6MP efficacy? How about is time x DNA-TGN level during maintenance therapy?

In the study, the authors identified 4 different heterozygotes and 1 compound heterozygote, but no homozygote, as well as the wild type genotype occupying the majority of the study subjects.

The readers may want to know the relationship between the variant genotypes and the results of 6MP-metabolites.

Do authors think that the results of DNA-TGN can be reflected to 6MP dosing at clinic immediately?

It takes at least two weeks after initiating 6MP administration. If so, it seems not practical.

Some minor comments are indicated below.

1. Patients with relapsed ALL are included in the study subjects.

Some relapsed patients may have received maintenance therapy with 6MP previously.

The episode could affect 6MP metabolism in somatic cells as well as leukemia cells.

2. Please clearly describe the definition of 6MP intensity. Is that the ration of each tolerable dose for 50mg/m2/day or a cumulative dose administrated before DNA-TGN measurement for a dosage calculated by 50mg/m2/day x administration days?

3. What is reasons for sex difference among the subjects?

How is the age distribution or the ratio of T-cell type malignancy?

Can authors have some discussion about sex difference in 6MP-tolerance or sensitivity?

4. One of purposes of this study was described as “to evaluate whether DNA-TGN can reflect the clinical changes during ALL treatment”. Median follow-up among the subjects was only 18.1 months. It is too short to evaluate clinical impact of the markers related to 6MP metabolism. If the study can include some surrogate markers such as change of MRD, this study could have more suggestive for clinical practice.

5. Is there any relationship between the average or minimum WBC count and DNA-TGN in patients with NUDT15 variants?

6. In Figure 1, a few patients with NUDT15 genetic variants showed more than 1.0 of 6MP intensity. What does it mean?

Reviewer #2: congratulations on this interesting work titled " DNA-thioguanine nucleotide as a treatment marker in acute lymphoblastic leukaemia patients with NUDT15 variant genotypes"

however, some issues in this manuscript need to be addressed before it is made available in the public domain.

Major

1) Difficult to follow the language

1) line 74: Inherited genetic variations in the anti-leukemic drug metabolism have been recognized as the risk factors for ALL relapse—may be authors meant toxicity as we know that we have sufficient literature on the evidence related to the incidence of

Citations for higher frequencies in Asians whether authors meant all Asians or specific populations within Asians- Japanese, Chinese, Indians, any other reports? Or public database where these frequencies are found

The study s main objective is to investigate the toxicity and introduction is focused on relapse?

Main research question seems association of DNA TGN levels with that of NUDT15 variants and consequent toxicities—one main problem with the methodology is that all the pati8ents were started with low dose which is however is an advantage for the investigation as they can compare the DNA TGN levels at low doses in patients with NUDT15 variants and without the variants.

More details on the treatment can be provided in supplementary material.

Instead of adjusting the tGN-DNA levels with 6MP intensity which is a ratio of 6MP dose given to that of standard dose. Authors should directly adjust the levels with the dose administered (normalization). Ideally higher TGN-DNA adjusted to the dose would be

LC-MS method validation is not described. Whether external validation QC samples or interday and intraday CV, as the study period is 2 years. – sometime the observed variability could be incorporated by the analytical method used, differences in the storage periods etc. Which sample was used how it was collected? not clear for the reader the main analyte measured to be associated with genotypes.

The ratio of DNA TGN/ TGN or DNA-TGN adjusted t that of 6MP dose ca be compared between the genotype groups using parametric or non-parametric tests depending upon the distribution of the data

Relapse and death are phenotypes that are too early to be evaluated during the study period.

Whether the time point of sample collection was associated with that of the levels ? as I see that authors must compare within each phase rather comparing distribution across all phases. For e.g. comparison within consolidation phase ? seems there is quite difference in the time points used for sample collection, why not standardizing the time point for sample collection.

Whether authors have used ROC analysis to define cutoffs for the DNA TGN levels associated with that of toxicity?

For inter individual variability, since authors have multiple readings for DNA TGN why not using repeated measured ANOVA (if data is normal distributed) or non-parametric tests (nonparametric marginal model) to compare the levels between genotype groups?

It was not clear if the authors investigated only specific varinats in NUDt15 or sequenced the whole?, if only looked at specific variants then variability may be explained by the presence or absence of other variants ?

Conclusion is not supported by the findings in the study. In fact authors showed the relation of DNA TGN levels and NUDt15 that correlated with that of 6MP dosing , because similar TGN DNA levels obtained using the low doses in variant carriers of NUDT15. IN this case, as authors warrant may be close monitoring of the individuals based on NUDT15 genotype ? for e.g. higher variability in levels seen in the variant carriers, so may be NUDt15 variant carriers need close monitoring of the levels and dose adjustments. However, variants were associated with that of 6MP dosing.

minor and when analysis is modified may be need ot modify the following .

Figure 1 is not easily readable. I would simply take the dose adjusted TGN DNA levels and compare between the groups based on genotypes. And then see the proportion of cytopenia based on ROC curve for these adjusted DNA TGN levels.

Fig 3A—the representation can be mentioned as the dose ratio i.e. last dose to the initial dose, and then compare the dose adjustment between the groups based on genotypes, that can give clear idea whether there was dose reduction or dose increment occurred within each group. —on parametric comparisons can be made as it is likely that the distribution would be non-normal

Fig 3B 6MP intensity can be shown as mean with SD bars on each cycle of maintenance-

Fig 3C also same as Fig 3B , Fig 3C is not readable and representation must be changed

Reviewer #3: Statistical Comments:

The statistical analysis part was collectively written, results are shown in Table(1),

with P-values where you have Three groups, all t-test, U-test are only for two groups

please clear that

In addition where is the results for Mixed Model

Please Pinpoint the statistical methods used for each P-value shown, so the

results will be more informative.

Reviewer #4: The authors reported a prospective study of serial collection of DNA-TGN in 72 patients including their NUDT15 genotypes and found that patients with NUDT15 variant showed higher DNA-TGN variability. The results are informative, however, there are few comments listed below to improve clarity.

Major comments

1. Table 1

There are eight patients genotyped as NUDT15 *1/*2 in this study, however, ref 14 had mentioned that the genotyping methods you used could not distinguish NUDT15 *1/*2 from *3/*6. Did the NUDT15 diplotypes for those patients had been confirmed by other methods?

2. Table 1

The statistical significance test (Chi-square test) used in table 1 was not suitable due to more than 20% of expected counts less than 5.

3. page 12, line 232-239

Please revise figure 2 by orders of these results you mentioned in manuscript and label figures (2A, 2B, 2C) after each results.

4. Line 257

Please correct the range of DNA-TGN.

5. line 259-260

Please correct the patient number and the range of DNA-TGN.

6. Table 2

Please add the unit used for measurements including DNA-TGN, RBC-TGN, MMPN, WBC and ANC. Also add measurement method for MMPN in “sample collection and analysis” section since MMPN had been mentioned here.

7. Discussion line 403-407

Please make more discussion and cite references to support your proposal since that the outcomes were not correlate with DNA-TGN level in this study.

8. Clinically, the NUDT15 genotyping will be done before the administration of 6MP. DNA-TGA will be available after the patients take the 6MP. The clinicans might adjust the dose of 6MP by white cell counts and ANC. Might you comment about this?

Minor comments

Pleases add p-value in figure 2 and figure 3A

6. PLOS authors have the option to publish the peer review history of their article (what does this mean?). If published, this will include your full peer review and any attached files.

Reviewer #1: No

Reviewer #2: No

Reviewer #3: No

Reviewer #4: No

---

## [Author Response · Author response to Decision Letter 0]

12 Nov 2020

< Response to editors and reviewers >

First of all, thank you for reviewing this paper in detail and giving valuable advices. We have made our best efforts to revise the manuscript, and we hope that it has been answered properly. Please let us know if there is lacking points. We will make additional corrections. We hope that the revised manuscript could be suitable for publication in PLOS ONE. 

(Please look at the separate file named "DNATGN_response to reviewers_last", as the text written on the web are easily scattered and figures/tables are not visible. )

< Editors’ comments >

- We have checked that this paper meets the style standards suggested by PLOS ONE. 

•The name of the colleague or the details of the professional service that edited your manuscript

- We have got the professional editing service by Editage.

•A copy of your manuscript showing your changes by either highlighting them or using track changes (uploaded as a *supporting information* file)

- We made a “DNATGN_revised manuscript with track changes_last” file. 

•A clean copy of the edited manuscript (uploaded as the new *manuscript* file)

 - We also made a “DNATGN_revised manuscript” file. 

3. You indicated that you had ethical approval and informed consent for your study.

In your Methods section, please ensure you have also stated whether you obtained consent from parents or guardians of the minors included in the study or whether the research ethics committee or IRB specifically waived the need for their consent.

 - Thank you for your comment. We additionally described the information in the methods section. 

 - Consent was obtained from parents for minors and from patients for adults. (line 152-153)

4. Please provide a sample size and power calculation in the Methods, or discuss the reasons for not performing one before study initiation.

 - Thank you for your comment. We discussed the reason for not performing sample size calculation in the methods section. 

 - The number of participants in the study was not set in advance because it was an exploratory study of DNA-TGN in patients treated according to the existing treatment schedule. (line 147-149) 

5. In your Methods section, please provide additional information about the participant recruitment method and the demographic details of your participants.

Please ensure you have provided sufficient details to replicate the analyses such as:

a) a description of any inclusion/exclusion criteria that were applied to participant recruitment,

- We appreciate for your comments. We described the content as follows;

- Patients who were not expected to receive 6MP treatment and those who did not wish to participate in the study were excluded from the study. (line 145-147)

b) the methods used to collect patient samples.

- We appreciate for your comments. We described the content as follows;

 - Prospective metabolite samples (DNA-TGN, erythrocyte 6-thioguanine nucleotides (RBC TGN), methylmercaptopurine nucleotides (MMPN)) were acquired simultaneously after administration of 6MP for at least 14 days after initiation of constant-dosing 6-MP treatment—at 2 weeks for the protocol taking 6MP for 2 weeks, at 2 and 4 weeks for the protocol taking 6MP for 4 weeks, at 2 and 4 and 8 weeks for the protocol taking 6MP for 8 weeks, and at 2 and 4 weeks of 1st cycle of maintenance treatment. The blood sampling for DNA-TGN was done along with routine samplings to monitor complete blood cell count and chemistry tests during chemotherapy. (line 168-175) 

6. Please ensure your methods are described in sufficient detail for others to replicate the study. Specifically, please provide a brief summary of the methods involved in LC-MS and genotyping.

- We have further described the contents you mentioned as follows.

- DNA- and RBC- TGN concentrations and MMPN were measured using liquid chromatography-tandem mass spectrometry (LC-MS). DNA-TGN test was done with isotope-labeled TG-d3 and guanine-d3 as internal standards as previously described. Chromatographic etheno-TG peaks were normalized using etheno-G by calculating TG responses as etheno-TG area/etheno-G area (DNA-TG=[etheno-TG response/etheno-G response]/[etheno-TG-d3 response/etheno-G-d3 response]).[12,13] (line 176-181) 

Genotyping of NUDT15 and TPMT was performed before the 6MP maintenance therapy by direct sequencing. Direct Sanger sequencing of exons 1 and 3 of NUDT15, and direct sequencing of exon 2 to 9 of TPMT were performed after PCR. [14,15]. (line 185-187) 

7. PLOS requires an ORCID iD for the corresponding author in Editorial Manager on papers submitted after December 6th, 2016. Please ensure that you have an ORCID iD and that it is validated in Editorial Manager. To do this, go to ‘Update my Information’ (in the upper left-hand corner of the main menu), and click on the Fetch/Validate link next to the ORCID field. This will take you to the ORCID site and allow you to create a new iD or authenticate a pre-existing iD in Editorial Manager. Please see the following video for instructions on linking an ORCID iD to your Editorial Manager account: https://www.youtube.com/watch?v=_xcclfuvtxQ

- Thank you for your comment. I validated the ORCID ID following the comment. 

8. Thank you for stating the following in the Financial Disclosure section:

'This study was supported by a grant from the Korean Foundation for Cancer Research (KFCR-2017-D-1). The funders had no role in study design, data collection and analysis, decision to publish, or preparation of the manuscript.'

We note that one or more of the authors are employed by a commercial company: Green Cross Laboratories

- The author who works at the Green Cross Laboratory previously worked at Samsung Medical Center with other authors. The Korean Foundation for Cancer Research, which provided funding for this study, is a completely separate institution from the workplace where the author above works. The funders did not participate in the design or implementation of the study, and only provided the research fund. 

- We also updated the author roles of the online submission form. 

-We additionally described following statement at the funding statement part. 

-The funder provided financial support for the study, but did not have any additional role in the study design, data collection and analysis, decision to publish, or preparation of the manuscript. The specific roles of the authors are articulated in the ‘author contributions’ section. (line 502-505)

- The commercial affiliation of Choi R, one of the co-authors, did not play a role in this study. 

- In the current state of revision, it is confirmed that all authors of this study have no conflict of interest.

We additionally described the following statement in the “Competing Interests Statement” part. 

- All authors have no potential conﬂicts of interest to declare. There is an author who belong to a commercial affiliation, but the commercial affiliation does not alter our adherence to PLOS ONE policies on sharing data and materials. (line 498-500) 

- We included both an updated Funding Statement and Competing Interests Statement in the cover letter.

Editor's comments:

As appended below, the reviewers have raised major concern/critique (Reviewer # 1 is against publication) and suggested further justification/work to consolidate the findings. Do go through the comments and amend the MS accordingly. What is the novel aspect of the present study? This MUST be stated clearly in the text. After incorporating all the amendments the MS should be checked by native speaker for grammar and syntax errors.

- Thank you for your kind comments. We stated the novel point of this study additionally in the introduction part. 

- To date, two papers have been published on DNA-TGN measured in patients with NUDT15 variant genotype. Previous studies have reported higher DNA-TGN level and toxicity in patients with the NUDT15 mutant genotype when 6MP is given in regular dose. In contrast, our study focused on DNA-TGN in patients with NUDT15 or TPMT variant genotypes, who were treated with reduced doses of 6MP. In addition, in this study, since repeated blood tests were performed in patients continuously receiving 6MP, changes in DNA-TGN over time could be observed. (line 127-133)

- Also, we modified the last paragraph of introduction to make the meaning clearer. 

 �- In this study, we investigated the pattern of DNA-TGN in patients harbouring the NUDT15 variants when 6MP was administered at reduced doses (dynamically adjusted based on degree of cytopenia). This study was designed to evaluate whether the DNA-TGN concentration differ based on genotype of NUDT15, and whether DNA-TGN can reflect the treatment-related toxicity during ALL treatment. In addition, we sought to determine the pattern of change in DNA-TGN during maintenance treatment including 6MP. (line 134-139)

< Reviewers’ comments >

Reviewer #1: The article investigated the amount of thioguanine incorporated into DNA (DNA-TGN) of mononuclear cells in patients receiving maintenance chemotherapy for acute lymphoblastic leukemia, and emphasized the importance of DNA-TGN monitoring in assessing the toxicity and efficacy of 6-mercaptopurine among those harboring NUDT15 variants.

The large parts of results are already reported in the previous reports, but they found high variability in DNA-TGN in patients with NUDT15 variant genotypes, which possibly lead to negative impacts on the outcome of leukemia.

The author concluded that DNA-TGN monitoring may be superior to the combination of the genomic diagnosis of NUDT15 genomic variants and 6MP dosing based on the results of genomic analysis for NUDT15 and TPMT. The conclusion is not completely proved with the results of this study.

For example, is the amount of DNA-TGN at one point sometime after 6MP-administration enough to assess the 6MP efficacy? How about is time x DNA-TGN level during maintenance therapy?

- Thank you for the kind advice. It would be ideal to access the time x DNA-TGN level as the reviewer suggested. But it was difficult to analyze the time x DNA-TGN level because there was no accurate record of the actual time taken and the time the DNA TGN was collected. However, it was assumed that there would be a certain level of regularity, as most patients took the drug on an empty stomach before bedtime (9-10p) and collected the blood sample at morning before clinic. In the future, while conducting prospective research, we will try to study on the advice given.

In the study, the authors identified 4 different heterozygotes and 1 compound heterozygote, but no homozygote, as well as the wild type genotype occupying the majority of the study subjects. The readers may want to know the relationship between the variant genotypes and the results of 6MP-metabolites.

- Thank you for the important point. We additionally analyzed 6MP metabolite results according to variant genotypes, and added the result as Figure 1B.

Figure 1B. DNA-TGN/6MP intensity ratio and rate of cytopenia according to pharmacogenetic subgroups. (Line presents median with interquartile range.)



Do authors think that the results of DNA-TGN can be reflected to 6MP dosing at clinic immediately? It takes at least two weeks after initiating 6MP administration. If so, it seems not practical.

- As the maintenance treatment lasts for more than 18 to 30 months, 6MP dosing based on genotypes and adjusting after administration at 2 or more weeks can help concise dosing afterwards for the long treatment period. 

- In this study, samples were collected prospectively but were analyzed after collection of sufficient samples to undergo LC-MS test for DNA-TGN. Thus, we could not change the dose according to DNA-TGN level.

- We added the reference for the method of this study, which was published during the review period of this article. (Reference number 13).

Some minor comments are indicated below.

1. Patients with relapsed ALL are included in the study subjects. Some relapsed patients may have received maintenance therapy with 6MP previously. The episode could affect 6MP metabolism in somatic cells as well as leukemia cells.

- Thank you for the attentive advice. In our study, there were 3 patients enrolled after relapse. The three patients had different genotypes: one patient with NUDT15 variant, another patient with TPMT variant, and a patient with wild type of both genes. Because the proportion of relapsed patients was small among the total, it was difficult to observe different metabolism in relapsed patients through our study. However, it was mentioned in the discussion that drug metabolism may change in relapsed patients. Also, we added a reference about this content. (Blood 2020;135:41-55.)

Since drug metabolism may differ in relapsed patients, it would be better to analyze relapsed patients separately if the number of patients is sufficient. [22] (line 451-452) 

2. Please clearly describe the definition of 6MP intensity. Is that the ration of each tolerable dose for 50mg/m2/day or a cumulative dose administrated before DNA-TGN measurement for a dosage calculated by 50mg/m2/day x administration days?

- Thank you for your comment. 6MP intensity means the explanation of the former. The description has been revised as follows.

- 6MP intensity was defined as the ratio between the prescribed daily 6MP dose (mg/m2/day) and the standard daily dose (50 mg/m2/day). (line 164-165)

3. What is reasons for sex difference among the subjects? How is the age distribution or the ratio of T-cell type malignancy? Can authors have some discussion about sex difference in 6MP-tolerance or sensitivity?

- We don't think there may have been any specific cause for the sex difference. The incidence of ALL is usually a bit higher in boys, and it is likely that the proportion of males who visited the hospital at the time of the study was particularly high.

- As advisement, the detailed diagnosis name including the T-/B-cell type of the patients and the age distribution at the time of enrollment are additionally listed in Table 1. (3rd and 4th row of Table 1) (line 235)

Additional analysis was performed to determine whether there was a difference according to sex, disease, and previous recurrence. 

- Kruskal-Wallis test was done to compare the difference of DNA-TGN according to sex, diagnosis and relapse state. (line 204-205) 

- When comparing the distribution of DNA-TGN, male (vs female, P=0.001) and Pre-B ALL (vs Pre-T ALL, lymphoma, P <0.001) showed higher level of DNA-TGN. However, when comparing DNA-TGN/6MP intensity, there was no significant difference according to sex (P=0.18), disease (P=0.061), and previous relapse (P=0.675). (line 254-257)

- We also added comments about sex difference in 6MP-tolerance. 

- Previous studies have reported that there is a difference in 6MP tolerance according to sex [21], but this study did not show a difference in DNA-TGN values compared to 6MP intensity according to sex. (line 437-439)

4. One of purposes of this study was described as “to evaluate whether DNA-TGN can reflect the clinical changes during ALL treatment”. Median follow-up among the subjects was only 18.1 months. It is too short to evaluate clinical impact of the markers related to 6MP metabolism. If the study can include some surrogate markers such as change of MRD, this study could have more suggestive for clinical practice.

- This is a very accurate point. As you mentioned, the follow-up period of our study was short, making it difficult to observe relapse or death. Since most of the patients had already acquired MRD negative at the time of DNA-TGN measurement, it was difficult to use MRD as well for clinical impact of markers.

- Clinical practice was mainly referring to treatment-related toxicity and 6MP dose adjustment, but the expression seemed to be unclear. To clarify the meaning, we changed the 1st paragraph of introduction as follows.

- Mercaptopurine is a long-term administered, important drug for treatment of childhood acute lymphoblastic leukemia (ALL). However, some patients experience toxicity during mercaptopurine treatment, such as severe infections related to bone marrow suppression, need of dose reduction, or frequent treatment interruptions. Mercaptopurine is known to have narrow therapeutic indices and it is also known that frequent dose change or discontinuation of mercaptopurine is associated with increased relapse of ALL. [1] Maintaining proper concentration of thiopurine metabolites is important for preventing the occurrence of relapse or toxic events after ALL treatment. (line 95-102)

5. Is there any relationship between the average or minimum WBC count and DNA-TGN in patients with NUDT15 variants?

- Thank you for your kind comment. The correlation between DNA-TGN and WBC was analyzed in patients with NUDT15 variants, and the result is further described.

 �- When separately analyzed the patients harbouring NUDT15 variant, the DNA-TGN level also showed no association with WBC. (P=0.87) (Line 314-315)

6. In Figure 1, a few patients with NUDT15 genetic variants showed more than 1.0 of 6MP intensity. What does it mean?

 - In some cases, starting with a small 6MP dose and being tolerable, the 6MP dose was administered with higher dose than the target at the later part of treatment. In the Methods section, we described the treatment in more detail.

- There were 5 patients who were tolerable to the drug and took higher than the intended dose at the end of the treatment. (Line 349-35)

Reviewer #2: Congratulations on this interesting work titled " DNA-thioguanine nucleotide as a treatment marker in acute lymphoblastic leukaemia patients with NUDT15 variant genotypes" However, some issues in this manuscript need to be addressed before it is made available in the public domain.

Major

1. Difficult to follow the language

1) line 74: Inherited genetic variations in the anti-leukemic drug metabolism have been recognized as the risk factors for ALL relapse—may be authors meant toxicity as we know that we have sufficient literature on the evidence related to the incidence of

The study s main objective is to investigate the toxicity and introduction is focused on relapse?

- Thank you for the clear comment. To clarify the meaning, we changed the 1st paragraph of introduction as follows.

- Mercaptopurine is a long-term administered, important drug for treatment of childhood acute lymphoblastic leukemia (ALL). However, some patients experience toxicity during mercaptopurine treatment, such as severe infections related to bone marrow suppression, need of dose reduction, or frequent treatment interruptions. Mercaptopurine is known to have narrow therapeutic indices and it is also known that frequent dose change or discontinuation of mercaptopurine is associated with increased relapse of ALL. [1] Maintaining proper concentration of thiopurine metabolites is important for preventing the occurrence of relapse or toxic events after ALL treatment. (line 95-102)

Citations for higher frequencies in Asians whether authors meant all Asians or specific populations within Asians- Japanese, Chinese, Indians, any other reports? Or public database where these frequencies are found

- Thank you for your comment. We added citations about NUDT15 variants in Asian population.

- Particularly, NUDT15 variants have been shown to be more prevalent in Asian populations than in other ethnic groups. [8-10] (line 121-122)

Main research question seems association of DNA TGN levels with that of NUDT15 variants and consequent toxicities—one main problem with the methodology is that all the patients were started with low dose which is however is an advantage for the investigation as they can compare the DNA TGN levels at low doses in patients with NUDT15 variants and without the variants.

- We agree with what you pointed out. Starting 6MP from a low dose, toxicity was rarely observed. However it was confirmed that DNA-TGN appeared differently depending on the NUDT15 status. We added some discussion about this point. 

- In this study, DNA-TGN test was performed at the beginning of the maintenance period in many cases. At this period, 6MP was administered at a low dose, and therefore, the frequency of cytopenia was very small. (line 433-435) 

More details on the treatment can be provided in supplementary material. 

Instead of adjusting the TGN-DNA levels with 6MP intensity which is a ratio of 6MP dose given to that of standard dose. Authors should directly adjust the levels with the dose administered (normalization). Ideally higher TGN-DNA adjusted to the dose would be

- Thank you for your thoughtful advice. As you mentioned, for standardization, it is better to apply the actual 6MP dose per body surface area, not 6MP intensity. 

- However, it is believed that DNA-TGN itself is a direct indicator of the actual therapeutic effect. To date, several papers published on the aspects of DNA-TGN in pediatric ALL have shown the value of DNA-TGN itself. In this study, we looked at the ratio because we thought that this would be a good index to see differences between groups, rather than the high meaning of the ratio itself.

LC-MS method validation is not described. Whether external validation QC samples or interday and intraday CV, as the study period is 2 years. – sometime the observed variability could be incorporated by the analytical method used, differences in the storage periods etc. Which sample was used how it was collected? not clear for the reader the main analyte measured to be associated with genotypes.

- In this study, samples were collected prospectively but were analyzed after collection of sufficient samples to undergo LC-MS test for DNA-TGN. 

- We added the reference for the method of this study, which was published during the review period of this article. (Reference number 13)

- Also, we described more about LC-MS method. 

- DNA-TGN test was done with isotope-labeled TG-d3 and guanine-d3 as internal standards as previously described. Chromatographic etheno-TG peaks were normalized using etheno-G by calculating TG responses as etheno-TG area/etheno-G area (DNA-TG=[etheno-TG response/etheno-G response]/[etheno-TG-d3 response/etheno-G-d3 response]).[12,13] (line 178-181)

The ratio of DNA TGN/ TGN or DNA-TGN adjusted t that of 6MP dose ca be compared between the genotype groups using parametric or non-parametric tests depending upon the distribution of the data. 

Whether authors have used ROC analysis to define cutoffs for the DNA TGN levels associated with that of toxicity?

Figure 1 is not easily readable. I would simply take the dose adjusted TGN DNA levels and compare between the groups based on genotypes. And then see the proportion of cytopenia based on ROC curve for these adjusted DNA TGN levels.

- The ratio of DNA-TGN/6MP intensity was compared between genotype groups using Mann-Whitney test. (Not normal distribution) P values are presented in a new figure, Figure 1B. 

- The rate of cytopenia was low, and rate of cytopenia did not show correlation with the genotypes. So, we could not draw the ROC curve. Instead, we tried to show the rate of cytopenia according to groups. (Figure 1B)

- Also, as there was also a request from other reviewer to show the distribution of DNA-TGN/6MP intensity ratio according to each subgroup, the groups were divided in this new figure. 

- Fig 1. (B) DNA-TGN/6MP intensity ratio and rate of cytopenia according to pharmacogenetic subgroups. (Line presents median with interquartile range.)

Relapse and death are phenotypes that are too early to be evaluated during the study period.

- This is a very accurate point. As you mentioned, the follow-up period of our study was short, making it difficult to observe relapse or death. Clinical respects were mainly referring to treatment-related toxicity and 6MP dose adjustment, but the expression seemed to be unclear in the manuscript. 

- At the beginning of the article, the part that mentioned about the relapse of ALL was deleted, and drug-related toxicity and difficulty in drug dose decision were emphasized.

- Mercaptopurine is a long-term administered, important drug for treatment of childhood acute lymphoblastic leukemia (ALL). However, some patients experience toxicity during mercaptopurine treatment, such as severe infections related to bone marrow suppression, need of dose reduction, or frequent treatment interruptions. Mercaptopurine is known to have narrow therapeutic indices and it is also known that frequent dose change or discontinuation of mercaptopurine is associated with increased relapse of ALL. [1] Maintaining proper concentration of thiopurine metabolites is important for preventing the occurrence of relapse or toxic events after ALL treatment. (line 95-102)

Whether the time point of sample collection was associated with that of the levels? as I see that authors must compare within each phase rather comparing distribution across all phases. For e.g. comparison within consolidation phase? seems there is quite difference in the time points used for sample collection, why not standardizing the time point for sample collection.

- Thank you for your careful advice. As you mentioned, we thought that there is a concern that the value of DNA-TGN will differ depending on the treatment schedule. In addition, the chemotherapeutic agents and doses were different for each patient in the treatment stage before maintenance treatment (from consolidation to delayed intensification), as each patient had different risk group and different type of leukemia or lymphoma. As a result, it was complicated to analyze levels separately for each protocol. Therefore, when looking at the DNA-TGN distribution by 6MP dose, analysis was performed only on the samples collected during the maintenance period.

- It is further described so that the content of this analysis can be clarified.

- For the association of DNA-TGN by 6MP dose and hematologic toxicity, analysis was performed only on the samples collected during maintenance treatment to rule out effects of other drugs. (line 199-201) 

For inter individual variability, since authors have multiple readings for DNA TGN why not using repeated measured ANOVA (if data is normal distributed) or non-parametric tests (nonparametric marginal model) to compare the levels between genotype groups?

- In fact, data were collected with the intention of performing ANOVA, but the timing of blood collection was not completely consistent for each patient. (For example, 25% of patients collected blood at 2,4,8 weeks, 20% at 2,8 weeks, 10% at 2,4,16 weeks, etc.) Therefore, we conducted a linear mixed model analysis, which we thought was the best method of analysis. This was described in the manuscript as next paragraph. Thank you for the comment.

- Linear mixed model analysis was performed to compare the pattern of change between NUDT15 variant and wild type groups. (line 2018-210)

- However, the linear mixed model analysis did not reveal any significant differences in the pattern of DNA-TGN concentration with respect to time, between patients harbouring NUDT15 wild type and variants (P=0.238). (line 352-354)

It was not clear if the authors investigated only specific variants in NUDT15 or sequenced the whole? if only looked at specific variants then variability may be explained by the presence or absence of other variants ?

- Thank you for your question. Previously, only references were provided for the test method and the contents were not written. We modified the manuscript to further describe the test method. 

- Direct Sanger sequencing of exons 1 and 3 of NUDT15, and direct sequencing of exon 2 to 9 of TPMT were performed after PCR. [14,15]. (line 186-187) 

Conclusion is not supported by the findings in the study. In fact authors showed the relation of DNA TGN levels and NUDt15 that correlated with that of 6MP dosing, because similar TGN DNA levels obtained using the low doses in variant carriers of NUDT15. IN this case, as authors warrant may be close monitoring of the individuals based on NUDT15 genotype? for e.g. higher variability in levels seen in the variant carriers, so may be NUDT15 variant carriers need close monitoring of the levels and dose adjustments. However, variants were associated with that of 6MP dosing.

- Thank you for your careful advice. As your comment, the fact that similar DNA-TGN levels were obtained with low dose of 6MP in patients with NUDT15 variant genotype is the most important point of the study. But as there were two previous articles which covered similar contents (although the 2 studies were done in small number of patients or animals), we wanted to show the new finding through this study. In our study, we thought that it was new to look at the changing pattern of DNA-TGN during the course of 6MP treatment. 

- It can be predicted that DNA-TGN variability may be higher in patients with NUDT15 variant because of the 6MP dose change. 

- However, the ratio of DNA-TGN divided by 6MP intensity was also higher in patients with NUDT15 variant, indicating that the variation in DNA-TGN was greater than from the dose changes. 

 - We wrote this explanation at line 352-330. 

 - This large variation in DNA-TGN and RBC TGN may be due to frequent changes in dose in the NUDT15 variant group. Therefore, we evaluated the variation in DNA-TGN/6MP intensity to eliminate the effect of dose variation. The CV for the DNA-TGN/6MP intensity ratios ranged from 23.8% to 90.5% (median 45.4%) in patients harbouring the NUDT15 variants, which was significantly higher (P=0.019) than that in patients harbouring wildtype NUDT15 (5.6% to 71.8%; median 32.3%). (line 330-335)

minor and when analysis is modified may be needed to modify the following.

Fig 3A—the representation can be mentioned as the dose ratio i.e. last dose to the initial dose, and then compare the dose adjustment between the groups based on genotypes, that can give clear idea whether there was dose reduction or dose increment occurred within each group. —on parametric comparisons can be made as it is likely that the distribution would be non-normal

- We appreciate for your advice. We modified the figure as your comment, presenting the dose ratio of last dose to the initial dose of 6MP. As there were only 2 patients who were treated last dose of 6MP in TPMT variant group, we presented only NUDT15 WT and NUDT15 variant group. Dots present each patient, box present median with interquartile range, and bars mean minimum to maximum value. 



Fig 3B 6MP intensity can be shown as mean with SD bars on each cycle of maintenance-

Fig 3C also same as Fig 3B, Fig 3C is not readable and representation must be changed

-Thank you for your advice. We changed the graph as mean with SD bars for each group, and combined two graphs (3B, 3C) in one graph (3B). 

 - Fig 3. Changes in 6MP dosage according to NUDT15 status. (A) ratio of starting 6MP dose and last 6MP dose of maintenance treatment. (B) dose changes in patients harbouring wild type NUDT15 and NUDT15 variants. (C) changes in the DNA-TGN/6MP intensity ratios over the course of the maintenance treatment.

Reviewer #3: Statistical Comments:

The statistical analysis part was collectively written, results are shown in Table(1),

with P-values where you have Three groups, all t-test, U-test are only for two groups

please clear that

Please Pinpoint the statistical methods used for each P-value shown, so the results will be more informative.

 - We appreciate for your advice. As there were 3 groups, Fisher’s exact test was performed to know differences in sex, diagnosis, and relapse status according to genotype group. We changed manuscript as follows.

 - Descriptive analyses were conducted for age, sex, diagnosis, NUDT15/TPMT genotype, 6-mercaptopurine dose, DNA-TGN and RBC TGN. Fisher’s exact test was performed to see differences in sex, diagnosis, and relapse status according to genotypes. For the association of DNA-TGN by 6MP dose and hematologic toxicity, analysis was performed only on the samples collected during maintenance treatment to rule out effects of other drugs. Correlation analysis was performed to see the correlation of DNA-TGN by 6MP dose and WBC by DNA-TGN. Fisher’s exact test was used to determine the differences in the rate of toxicity in the NUDT15 variant and wild type groups. Kruskal-Wallis test was done to compare the difference of DNA-TGN according to sex, diagnosis and relapse state. Also, same test was used to compare the difference of DNA-TGN, 6MP intensity, DNA-TGN/6MP intensity according to genotypes. To evaluate the distribution of DNA-TGN, DNA-TGN/6MP intensity and RBC TGN values between and within patients, coefficients of variation (CV) were calculated. Linear mixed model analysis was performed to compare the pattern of change between NUDT15 variant and wild type groups. (methods, line 197-210)

 - In addition, where is the results for Mixed Model

 - The results are shown at line 344-346, as follows: 

 - However, the linear mixed model analysis did not reveal any significant differences in the pattern of DNA-TGN concentration with respect to time, between patients harbouring NUDT15 wild type and variants (P=0.238). (line 351-353)

Reviewer #4: 

The authors reported a prospective study of serial collection of DNA-TGN in 72 patients including their NUDT15 genotypes and found that patients with NUDT15 variant showed higher DNA-TGN variability. The results are informative, however, there are few comments listed below to improve clarity.

Major comments

1. Table 1

There are eight patients genotyped as NUDT15 *1/*2 in this study, however, ref 14 had mentioned that the genotyping methods you used could not distinguish NUDT15 *1/*2 from *3/*6. Did the NUDT15 diplotypes for those patients had been confirmed by other methods?

- Thank you for your comment. As you mentioned, NUDT15 genotype tests were performed in patients in this study in the same way as in Reference 14. Direct sequencing alone is not enough to distinguish *1/*2 and *3/*6, and additional tests such as parental tests were not carried out. So technically, it is difficult to distinguish between *1/*2 and *3/*6. We additionally described about this fact as follows. 

- Since NUDT15 *1/*2 and *3/*6 genotypes were not distinguished technically by our method, *1/*2 strictly represents *1/*2 or *3/*6 [14]. (line 187-189)

2. Table 1

The statistical significance test (Chi-square test) used in table 1 was not suitable due to more than 20% of expected counts less than 5.

- We appreciate for the kind review. (Initially, the groups were divided as 2 groups; NUDT15 variant and wild type. We may have used Chi-square test at that time, and did not carefully modified when changing groups. Thank you for the comment.) Instead of Chi-square test, Fisher’s exact test was performed to know differences in sex, diagnosis, and relapse status according to genotype group. We changed manuscript and table as follows. (line 198-199)

 - Fisher’s exact test was performed to see differences in sex, diagnosis, and relapse status according to genotypes. 

3. page 12, line 232-239

Please revise figure 2 by orders of these results you mentioned in manuscript and label figures (2A, 2B, 2C) after each result.

- We agree with what you pointed out. We reordered figure 2 and mentioned in manuscript as follows. (line 274-281) 

Patients with NUDT15 variant genotypes were treated with significantly lower 6MP intensities when compared to patients with NUDT15 wild type genotypes (P<0.001). (Fig 2A) However, there was no significant difference in the DNA-TGN levels between patients with NUDT15 variant genotypes, TPMT variant genotypes, and both of NUDT15 and TPMT wild types (P=0.261). (Fig 2B) The ratio of DNA-TGN/6MP intensity was significantly higher in patients harbouring NUDT15 variants than that in patients harbouring wild type NUDT15 (P<0.001), but no difference was observed between patients harbouring TPMT variant and wild type (P=0.743). (Fig 2C) (line 284-285)

- Fig 2. Distribution of various parameters based on genotype. (A) 6MP intensity (B) DNA-TGN (C) DNA-TGN/6MP intensity (WT; wild type, v; variant)

4. Line 257

Please correct the range of DNA-TGN.

 - We deeply appreciate for your detailed review. We corrected the range of DNA-TGN as follows.

(line 296-297)

 - The DNA-TGN concentrations during these leukopenia episodes ranged from 27.8 to 504.8 fmol TG/µg DNA. 

5. line 259-260

Please correct the patient number and the range of DNA-TGN.

- Thank you for your detailed review. We apologize for writing the contents incorrectly and corrected manuscript as follows.

(line 297-301)

 - Three of the episodes developed leukopenia before 12 weeks of treatment, and their 6MP intensity was 0.17, 0.12, and 0.08, respectively. However, in the five cases of leukopenia that occurred after 12 weeks of 6MP-based maintenance therapy, the 6MP intensity ranged from 0.22 to 0.98 (median 0.4).

6. Table 2

Please add the unit used for measurements including DNA-TGN, RBC-TGN, MMPN, WBC and ANC. Also add measurement method for MMPN in “sample collection and analysis” section since MMPN had been mentioned here.

- Thank you for your detailed review. We added the units which were missing as follows. (Table 2) 

(line 302)

DNA-TGN RBC TGN MMPN‡ WBC ANC

(fmol TG/µg DNA) (µmol/L) (µmol/L) (/µL) (/µL)

- We also add measurement method for MMPN in “sample collection and analysis” section, as commented. 

(line 168-171)

 - Prospective metabolite samples (DNA-TGN, erythrocyte 6-thioguanine nucleotides (RBC TGN), methylmercaptopurine nucleotides (MMPN)) were acquired simultaneously after administration of 6MP for at least 14 days after initiation of constant-dosing 6-MP treatment (line 176-177)

DNA- and RBC- TGN concentrations and MMPN were measured using liquid chromatography-tandem mass spectrometry (LC-MS). 

7. Discussion line 403-407

Please make more discussion and cite references to support your proposal since that the outcomes were not correlate with DNA-TGN level in this study.

 - Thank you for the advice. It seems that the existing sentence that you pointed out did not accurately express the point of the study, so the sentence was modified as follows. 

(line 458-463)

- Through this study, it was confirmed that patients with NUDT15 variant genotype could achieve a similar level of DNA-TGN even if they administered lower dose of 6MP, but subsequent dose adjustments by DNA-TGN were impossible because the DNA-TGN test was not processed in real time. We propose a larger study that starts reduced 6MP dose in patients harbouring the NUDT15 or TPMT variants, and then titrate the dose based on the DNA-TGN levels. 

- In addition, the reason why the clinical results were not consistent with DNA-TGN was mentioned more in the discussion 

- (line 433-435) In this study, a DNA-TGN test was often performed at the beginning of the maintenance period. At this time, 6MP was administered at a low dose, and therefore, the frequency of cytopenia was very small. 

8. Clinically, the NUDT15 genotyping will be done before the administration of 6MP. DNA-TGA will be available after the patients take the 6MP. The clinicians might adjust the dose of 6MP by white cell counts and ANC. Might you comment about this?

- We appreciate for your advice. We added a comment about the process in the manuscript.

- (line 181-185)

In this study, blood samples were collected prospectively, but the DNA-TGN test was not performed in real time. Therefore, the dose of 6MP was adjusted according to white blood cell (WBC) count, aspartate aminotransferase (AST), alanine transaminase (ALT), but not DNA-TGN values. 

Minor comments

Pleases add p-value in figure 2 and figure 3A

- Thank you for your advice. The order of Figure 2 was changed, following the advice. And we added the P value also. 

(line 284-285)

Fig 2. Distribution of various parameters based on genotype. (A) 6MP intensity (B) DNA-TGN (C) DNA-TGN/6MP intensity (WT; wild type, v; variant)

 Fig 3A was changed as presenting ratio of last 6MP dose vs initial dose according to groups, following the advice of another reviewer. And we added the P value also.

Fig 3. Changes in 6MP dosage according to NUDT15 status. (A) ratio of starting 6MP dose and last 6MP dose of maintenance treatment

---

## [Decision Letter · Decision Letter 1]

10 Dec 2020

PONE-D-20-27853R1

DNA-thioguanine nucleotide as a treatment marker in acute lymphoblastic leukaemia patients with NUDT15 variant genotypes

PLOS ONE

Dear Dr. Koo,

Thank you for submitting your manuscript to PLOS ONE. After careful consideration, we feel that it has merit but does not fully meet PLOS ONE’s publication criteria as it currently stands. Therefore, we invite you to submit a revised version of the manuscript that addresses the points raised during the review process.

ACADEMIC EDITOR: Still reviewers' are raising major concern over the revised form of the MS. Do go through the comments and amend the MS accordingly. Furthermore

1- Authors' qualifications (MD, PhD) are not needed, delete 

2- Justify the text throughout the MS

3- Line 75: Avoid using "will" as well as throughout the text

4- Add 2 more keywords

5- Periods shouldn't be before Ref. No. It should be after Ref, No. For instance: of mercaptopurine is associated with increased relapse of ALL. [1] SHOULD be of mercaptopurine is associated with increased relapse of ALL [1]. Revise throughout the text

6- Don't use "We, our". Use impersonal phrasing throughout the text

7- Line 177: liquid chromatography-tandem mass spectrometry (LC-MS) is not correct. This should be LC-MS/MS. Do you use LC-MS or LC-MS/MS? 

8- Line 177: Details regarding analysis, including method optimization and performance should be added as Suppl. materials

9- Study strength and limitations should be in a separate section headed as mentioned. It should be ahead of Conclusion. Start with strength followed by limitations

10- Conclusion should be in a separate section. What are the clinical relevance and future perspective? Add this to conclusion section

We look forward to receiving your revised manuscript.

Kind regards,

A. M. Abd El-Aty

Academic Editor

PLOS ONE

Reviewers' comments:

Reviewer's Responses to Questions

**Comments to the Author**

1. If the authors have adequately addressed your comments raised in a previous round of review and you feel that this manuscript is now acceptable for publication, you may indicate that here to bypass the “Comments to the Author” section, enter your conflict of interest statement in the “Confidential to Editor” section, and submit your "Accept" recommendation.

Reviewer #1: (No Response)

Reviewer #2: (No Response)

Reviewer #3: All comments have been addressed

Reviewer #4: All comments have been addressed

2. Is the manuscript technically sound, and do the data support the conclusions?

Reviewer #1: Partly

Reviewer #2: Partly

Reviewer #3: Yes

Reviewer #4: Yes

3. Has the statistical analysis been performed appropriately and rigorously? 

Reviewer #1: I Don't Know

Reviewer #2: No

Reviewer #3: Yes

Reviewer #4: Yes

4. Have the authors made all data underlying the findings in their manuscript fully available?

Reviewer #1: Yes

Reviewer #2: Yes

Reviewer #3: Yes

Reviewer #4: Yes

5. Is the manuscript presented in an intelligible fashion and written in standard English?

Reviewer #1: Yes

Reviewer #2: No

Reviewer #3: Yes

Reviewer #4: Yes

6. Review Comments to the Author

Reviewer #1: (Major comments)

The authors concluded that the dose adjustment of 6MP doses based on WBC count was not effective to maintain DNA-TGN constantly.

Can they exclude the cytotoxic effect of co-administered MTX?

They showed that DNA-TGN correlates positively with 6MP dose in those with wild type NUDT-15. Therefore, they can say the additional effect of MTX on WBC count may be minimal. But the size of cohort with NUDT-15 variants is much smaller than those with wild type NUDT-15. Can they show that patients with genetic variants related with MTX toxicity were not included in the cohort with NUDT-15 variants?

In discussion section, the authors speculated a reason of non-correlation between DNA-TGN or RBC-TGN and WBC. In this part, effect of MTX should be discussed.

Let me allow me give the similar question to the review for the first submission. The measurement of DNA-TGN was carried out LC-MS with radio-isotpoic reagents. They suggested the need for close DNA-TGN monitoring to allow for the development of more finely tuned treatment plan. Is it practicable for monitoring of individual patients coming to clinic weekly or biweekly?

DNA-TGN reflects the damage of DNA during the relatively short-time after 6MP-administration while RBC-TGN means the accumulation of 6MP metabolites in cytoplasm representing the relatively long-term effect of 6MP. Can the author discussed the difference of the two parameters and difference in the impact of the parameters on WBC count during 6MP-treatment?

(Minor comments)

Conclusion of Abstract. The last sentence ; will be important for what? Clinical outcome or adverse effects?

Page 5. L111; Erythrocyte → erythrocyte

Page 8. L171; at 2 weeks; at 2nd week (?)

Reviewer #2: introduction modified statements were not referenced appropriately

statistical analysis suggestions were not appropriately implemented , normalization of DNA TGN levels to the dose of 6MP , authors claim several papers here in this reply on DNA TGN levels, and in the introduction they say there are only two reports available- conflicting statements. Normalization is required when we wanted to see effect of genetic variant as dose changes by itself could change the levels of DNA TGN. so it has to be normalized to be able tp compare between genotype groups.

they were also not sure about the timing of the sample collecting sued for analysis-- this was not disclosed clearly

now the manuscript has been improved a lot , however need more precisions to accurately represent the data

Reviewer #3: (No Response)

Reviewer #4: There were several questions from different reviewers and the editor. All questions were answered properly. I have no other comments.

7. PLOS authors have the option to publish the peer review history of their article (what does this mean?). If published, this will include your full peer review and any attached files.

Reviewer #1: No

Reviewer #2: No

Reviewer #3: No

Reviewer #4: No

---

## [Author Response · Author response to Decision Letter 1]

15 Dec 2020

Response to reviewers (2nd) 

We sincerely appreciate the insights and constructive suggestions. We have made corrections based on the reviewer’s comment and suggestions. We sincerely hope that the manuscript is now suitable for publication in this journal and would be pleased to respond to any further queries regarding this submission. Thank you for your consideration.

Response to the academic editor 

1- Authors' qualifications (MD, PhD) are not needed, delete 

→Thank you for the comment. We have also removed qualifications from the title page.

2- Justify the text throughout the MS 

→ Thank you for the comment. We justified the text throughout the manuscript. 

3- Line 75: Avoid using "will" as well as throughout the text

 → Thank you for the advice which helps make the improved manuscript. We followed the advice.

4- Add 2 more keywords

 → Thank you for the comment. We added 2 more keywords. 

5- Periods shouldn't be before Ref. No. It should be after Ref, No. For instance: of mercaptopurine is associated with increased relapse of ALL. [1] SHOULD be of mercaptopurine is associated with increased relapse of ALL [1]. Revise throughout the text

 → Thank you for the advice which helps make the improved manuscript. We changed the format according to the advice. 

6- Don't use "We, our". Use impersonal phrasing throughout the text

→Thank you for the comment. We followed the advice.

7- Line 177: liquid chromatography-tandem mass spectrometry (LC-MS) is not correct. This should be LC-MS/MS. Do you use LC-MS or LC-MS/MS? 

 → Thank you for the detailed review. We changed the word according to the comment. 

8- Line 177: Details regarding analysis, including method optimization and performance should be added as Suppl. Materials on 

 → Thank you for your comment. The test method implemented by our center is described in detail in the paper dealing DNA-TGN test development, so the paper is cited. (reference number 17) If it is better to write the contents of the references as supplementary data, we will implement it.

9- Study strength and limitations should be in a separate section headed as mentioned. It should be ahead of Conclusion. Start with strength followed by limitations

 → Thank you for the advice. We changed the structure as recommended. 

10- Conclusion should be in a separate section. What are the clinical relevance and future perspective? Add this to conclusion section

→ Thank you for the constructive advice. We changed the structure and added the content to conclusion section. 

Response to the reviewers 

Reviewer #1: (Major comments)

The authors concluded that the dose adjustment of 6MP doses based on WBC count was not effective to maintain DNA-TGN constantly. Can they exclude the cytotoxic effect of co-administered MTX? They showed that DNA-TGN correlates positively with 6MP dose in those with wild type NUDT-15. Therefore, they can say the additional effect of MTX on WBC count may be minimal. But the size of cohort with NUDT-15 variants is much smaller than those with wild type NUDT-15. Can they show that patients with genetic variants related with MTX toxicity were not included in the cohort with NUDT-15 variants?

In discussion section, the authors speculated a reason of non-correlation between DNA-TGN or RBC-TGN and WBC. In this part, effect of MTX should be discussed.

→ The authors appreciate the comments.

→ About genetic variants related with MTX toxicity, genotyping of methylenetetrahydrofolate reductase (MTHFR) was done. Fifty-five of the 71 patients (77%) enrolled in the study were tested for MTHFR, but 16 patients (23%) did not undergo MTHFR study. Due to the large missing data, it was difficult to analyze the relationship between the MTHFR genotype and the leukocyte count. 

→ In addition, in this study, MTX was administered in full dose except in cases where toxicity was evident during maintenance treatment. As MTX was usually administered at a constant dose, it was thought that the degree of toxicity due to MTX would not be variable. Therefore, the degree of toxicity according to the MTX dose was not included in this study.

→ This background is further described in the discussion section.

→ (line 440-¬444) Since methotrexate (MTX) is continuously taken in addition to 6MP during maintenance treatment, treatment-related toxicity can be caused by both drugs. However, in this study, MTX was not included in the analysis because it was considered that there would be little variation in toxicity due to MTX since the dose of MTX was kept near constant in each patient. 

Let me allow me give the similar question to the review for the first submission. The measurement of DNA-TGN was carried out LC-MS with radio-isotpoic reagents. They suggested the need for close DNA-TGN monitoring to allow for the development of more finely tuned treatment plan. Is it practicable for monitoring of individual patients coming to clinic weekly or biweekly?

→ Thank you for the important advice. 

→ The DNA-TGN test was performed by the LC-MS/MS method using radio-isotope agents, but the substance was not injected into the patient. A technique using radio-isotope agents was used for the collected blood.

→ As the use of kits and materials for each test is burdensome for the research cost, in this study, the patient's blood was collected, prepared, and stored for testing. The test for DNA-TGN was performed when more than a certain number of samples were collected. Therefore, it took time to check the test results, making it difficult to adjust the treatment according to DNA-TGN test results. When DNA-TGN test is stably set up in the laboratory and cost for study is established, it is expected that DNA-TGN test will be possible to carry out frequently. In the future, it would be helpful to perform the DNA-TGN test at few weeks later from taking 6MP maintenance (when thought to be reached the plateau), when the 6MP dose is changed, or when toxicity occurs. 

→ These comments have been briefly added to the discussion section.

→ (line 430-432) Considering the timing of the toxicity onset, it would be helpful to test DNA-TGN after 2 weeks or more, when the dose of 6MP is changed, or when toxicity occurs, rather than immediately after starting 6MP. 

DNA-TGN reflects the damage of DNA during the relatively short-time after 6MP-administration while RBC-TGN means the accumulation of 6MP metabolites in cytoplasm representing the relatively long-term effect of 6MP. Can the author discussed the difference of the two parameters and difference in the impact of the parameters on WBC count during 6MP-treatment?

→ Thank you for the valuable comment. 

→ DNA-TGN is a final active metabolite of all the thiopurines, which is produced by incorporation of 6-TGN into DNA. 6-TGN accumulation in DNA is an inevitable consequence of thiopurine treatment and the central to the therapeutic effects. It is believed that the S phase of the cell cycle must pass once before 6TGN is incorporated DNA, then at least one more S phase to allow replication of the 6-TGN substituted DNA, resulting in cytotoxicity. 

→ According to previous studies, 6-TGN concentration reached a plateau after 2 weeks to several weeks after starting 6MP. Although there is no clear report when DNA-TGN reached its highest concentration after taking a fixed dose of 6MP, most studies performed sampling for DNA-TGN test at 2 weeks or more of 6MP administration. 

→ These characteristics and the appropriate timing of DNA TGN testing are further described in the discussion section.

→ (line 428-432) Thiopurine is known to present delayed cytotoxic effect. At least two or more S phase of cell cycles are needed for 6TGN to incorporate into DNA, and this 6TGN-substituted DNA replicates and present mismatch, eventually leading to cell death [4]. Considering the timing of the toxicity onset, it would be helpful to test DNA-TGN after 2 weeks or more, when the dose of 6MP is changed, or when toxicity occurs, rather than immediately after starting 6MP. 

(Minor comments)

Conclusion of Abstract. The last sentence; will be important for what? Clinical outcome or adverse effects?

→ Thank you for your comment. To clarify the expression, it has been modified as follows.

→ (line 74-76) Particularly in patients with NUDT15 variants who need to reduce the 6MP dose, DNA-TGN could be applied as a useful marker to monitor the therapeutic effect of 6MP. 

Page 5. L111; Erythrocyte → erythrocyte

→ Thank you for the detailed comment. We edited the word according to the comment. 

→ (line 110-112) Other studies have evaluated the use of erythrocyte TGN (RBC TGN) as an indicator of 6MP metabolism, but these studies have found that RBC TGN is not a robust prognostic marker for relapse [3]. 

Page 8. L171; at 2 weeks; at 2nd week (?)

→ Thank you for the comment. We edited the manuscript according to the comment. 

→ after 2 weeks for the protocol taking 6MP for 2 weeks, after 2 and 4 weeks for the protocol taking 6MP for 4 weeks, after 2 and 4 and 8 weeks for the protocol taking 6MP for 8 weeks, and after 2 and 4 weeks of 1st cycle of maintenance treatment. (line 170-173)

Reviewer #2: 

Introduction modified statements were not referenced appropriately 

→ Thank you for your careful advice. As your comment, some introduction statements were not referenced. We added the references as follows.

→ (line 117-118) Several genes have been linked to the metabolism of 6MP used in the treatment of ALL; this include TPMT and NUDT15 [6]. 

→ (line 127-128) To date, three papers have been published on DNA-TGN measured focusing in patients with NUDT15 variant genotype. [13-15] 

Statistical analysis suggestions were not appropriately implemented, normalization of DNA TGN levels to the dose of 6MP, Normalization is required when we wanted to see effect of genetic variant as dose changes by itself could change the levels of DNA TGN. So it has to be normalized to be able tp compare between genotype groups.

→ Thank you for valuable comments. I think I did not fully understand the meaning at first revision that the data needs normalization. At 2nd revision, normalization of DNA TGN levels to 6MP intensity was done according to the advice. 

→ When the analysis was done after normalization, the P values (which were not statistically significant) changed. Changes in the analysis method were described in the method section, and the changed statistical results were reflected in the manuscript and figure. 

→ (line 206-207) To see the difference in DNA-TGN/6MP intensity according to genotype, normalization of DNA-TGN/6MP intensity was done. 

→ (line 276-278) The ratio of DNA-TGN/6MP intensity was significantly higher in patients harbouring NUDT15 variants than that in patients harbouring wild type NUDT15 (P<0.001), but no difference was observed between patients harbouring TPMT variant and wild type (P=0.323). (Fig 2C) Figure 2C 

Authors claim several papers here in this reply on DNA TGN levels, and in the introduction they say there are only two reports available- conflicting statements. 

→ There have been reports of DNA-TGN levels in several papers, but there were only few papers reporting the DNA-TGN in patients with the NUDT15 variants. It seems that it was written inaccurately so that the meaning does not delivered well, so I modified it as follows. 

→ In addition, when we have reviewed the literature based on the reviewer’s suggestion, we found one more article that partially overlaps the previously found articles, so citation was added. Thank you for the attentive advice.

→ (line 127-130) To date, three papers have been published on DNA-TGN measured focusing in patients with NUDT15 variant genotype. [13-15] These studies have reported that patients with the NUDT15 variant genotype showed higher DNA-TGN level and toxicity when 6MP was given in regular dose. 

→ (line 403-406) Three previously published papers have demonstrated that higher DNA-TGN concentrations coincided with a higher incidence of cytopenia in patients harbouring the NUDT15 variants when the patients received the standard dose of 6MP [13-15]. 

They were also not sure about the timing of the sample collecting sued for analysis-- this was not disclosed clearly

→ Thank you for the important advice. We additionally described the conditions of the samples used in the analysis in more detail. 

→ (line 210-213) When performing CV and linear mixed model analyses to determine the degree of variability, only the values collected during the maintenance treatment of patients who took DNA-TGN samples more than three times during the maintenance period were used in the analysis. 

→ The sentence below was previously written, but the order of the sentences was arranged differently to better understand them.

→ (line 201-203) For the association of DNA-TGN by 6MP dose and hematologic toxicity, analysis was performed only on the samples collected during maintenance treatment to rule out effects of other drugs. 

Reviewer #3: 

(No Response)

→ We appreciate your evaluation.

Reviewer #4: 

There were several questions from different reviewers and the editor. All questions were answered properly. I have no other comments.

→ We sincerely appreciate the positive evaluation of the manuscript.

---

## [Decision Letter · Decision Letter 2]

29 Dec 2020

PONE-D-20-27853R2

DNA-thioguanine nucleotide as a treatment marker in acute lymphoblastic leukemia patients with *NUDT15* variant genotypes

PLOS ONE

Dear Dr. Koo,

Thank you for submitting your manuscript to PLOS ONE. After careful consideration, we feel that it has merit but does not fully meet PLOS ONE’s publication criteria as it currently stands. Therefore, we invite you to submit a revised version of the manuscript that addresses the points raised during the review process.

ACADEMIC EDITOR:

1- Again, don't use "Will" throughout the text

2- Don't use "We, our". Use impersonal phrasing throughout the text

3- Revise Section heading to "Study strength and limitations" instead of the current one

4- Again, comments (8) 8- Line 177: Details regarding analysis, including method optimization and performance should be added as Suppl. Materials

5- Please check grammar and syntax errors with the assistance from native speaker

We look forward to receiving your revised manuscript.

Kind regards,

A. M. Abd El-Aty

Academic Editor

PLOS ONE

Reviewers' comments:

Reviewer's Responses to Questions

**Comments to the Author**

1. If the authors have adequately addressed your comments raised in a previous round of review and you feel that this manuscript is now acceptable for publication, you may indicate that here to bypass the “Comments to the Author” section, enter your conflict of interest statement in the “Confidential to Editor” section, and submit your "Accept" recommendation.

Reviewer #1: All comments have been addressed

Reviewer #2: All comments have been addressed

2. Is the manuscript technically sound, and do the data support the conclusions?

Reviewer #1: Yes

Reviewer #2: Yes

3. Has the statistical analysis been performed appropriately and rigorously? 

Reviewer #1: I Don't Know

Reviewer #2: Yes

4. Have the authors made all data underlying the findings in their manuscript fully available?

Reviewer #1: Yes

Reviewer #2: No

5. Is the manuscript presented in an intelligible fashion and written in standard English?

Reviewer #1: Yes

Reviewer #2: Yes

6. Review Comments to the Author

Reviewer #1: (No Response)

Reviewer #2: (No Response)

7. PLOS authors have the option to publish the peer review history of their article (what does this mean?). If published, this will include your full peer review and any attached files.

Reviewer #1: No

Reviewer #2: No

---

## [Author Response · Author response to Decision Letter 2]

31 Dec 2020

Response to the academic editor 

We sincerely appreciate the detailed review with suggestions. We have made corrections based on the comments. We sincerely hope that the manuscript is now suitable for publication in this journal and would be pleased to respond to any further queries regarding this submission. Thank you for your consideration.

1- Again, don't use "Will" throughout the text

Thank you for the detailed comment. We removed “will” from the text.

- In addition to the few papers on DNA-TGN so far, this paper can (<- will) help establish the standard for therapeutic dose adjustment of 6MP based on DNA-TGN. (line 451) 

- In the future, larger studies are (<- will be) needed to reduce the 6MP dose in patients with NUDT15 or TPMT variants and then titrate the dose based on DNA-TGN levels. (line 469) 

2- Don't use "We, our". Use impersonal phrasing throughout the text

Thank you for the advice. We edited the sentences which included “we, our” as follows. 

- This study was able to demonstrate the efficacy of DNA-TGN in monitoring the treatment effects of 6MP, particularly in patients harbouring NUDT15 variant genotype. (line 481)

- This appears to be the first study to analyze prospective serial DNA-TGN samples in a large number of patients—including those, which carry NUDT15 variant genotypes—treated with reduced doses of 6MP. (line 455) 

3- Revise Section heading to "Study strength and limitations" instead of the current one

Thank you for the advice. We changed the heading to “The significance of the study”.

4- Again, comments (8) 8- Line 177: Details regarding analysis, including method optimization and performance should be added as Suppl. Materials

Thank you for the advice which helps make the improved manuscript. We added the details regarding sample analysis, including method optimization and performance as supplementary material. 

 - S1 Appendix. Detailed description of study methods. 

5- Please check grammar and syntax errors with the assistance from native speaker

- Thank you for the constructive advice. The manuscript underwent additional proofreading in English. English proofreading certificate is uploaded, also.

---

## [Editor Report · Decision Letter 3]

4 Jan 2021

PONE-D-20-27853R3

DNA-thioguanine nucleotide as a treatment marker in acute lymphoblastic leukemia patients with *NUDT15* variant genotypes

PLOS ONE

Dear Dr. Koo,

Thank you for submitting your manuscript to PLOS ONE. After careful consideration, we feel that it has merit but does not fully meet PLOS ONE’s publication criteria as it currently stands. Therefore, we invite you to submit a revised version of the manuscript that addresses the points raised during the review process.

ACADEMIC EDITOR: Again, authors' have to amend the MS very carefully based upon the comments stated below

1- This is the 3rd time authors' are asked to revise the last paragraph before conclusion to "Study strength and limitations". Unfortunately, this is not done, although they have said done. Heading MUST be changed to what has been stated and related text should be in one paragraph

2- Line 466-472: This is a redundancy for conclusion, avoid repetition

3- Where is the results of method validation? Linearity, LOD, LOQ, Recovery, and RSD values. This should be added in the Suppl. materials

4- Suppl. Materials should be checked as well for grammar and syntax errors

5- All amendments should be highlights yellow. Don't use track-changes mode

We look forward to receiving your revised manuscript.

Kind regards,

A. M. Abd El-Aty

Academic Editor

PLOS ONE

---

## [Author Response · Author response to Decision Letter 3]

5 Jan 2021

Response to the academic editor 

We sincerely appreciate the detailed review with suggestions. Also, we apologize for some misunderstandings and improper corrections. We have made corrections based on the comments again. We sincerely hope that the manuscript is now suitable for publication in PLOS ONE. It will be of great help if a quick decision is possible. Thank you for your consideration.

ACADEMIC EDITOR: Again, authors have to amend the MS very carefully based upon the comments stated below

1- This is the 3rd time authors' are asked to revise the last paragraph before conclusion to "Study strength and limitations". Unfortunately, this is not done, although they have said done. Heading MUST be changed to what has been stated and related text should be in one paragraph

Response: We apologize for the misunderstanding and improper correction. We changed the heading to “Study strength and limitations” as recommended, and made the text into one paragraph. 

2- Line 466-472: This is a redundancy for conclusion, avoid repetition

Response: We truly agree with your comment. As mentioned by the editor, we deleted the contents that overlap with the contents of the conclusion section.

3- Where is the results of method validation? Linearity, LOD, LOQ, Recovery, and RSD values. This should be added in the Suppl. Materials

Response: We do appreciate the detailed suggestion. We added the results of method validation for LC-MS/MS of DNA-TG quantification. In this study, LOB was experimentally confirmed, but LOD was not. (As two of the guidelines for LC-MS/MS method validation recommended LOD but was not recommended by other guidelines, LOD was not confirmed in this study.) Linearity, LOB, LLOQ, recovery, and RSD valued are presented. 

4- Suppl. Materials should be checked as well for grammar and syntax errors

Response: Thank you for the comment. Supplementary index was checked for grammar and syntax errors, also. 

5- All amendments should be highlights yellow. Don't use track-changes mode

Response: We appreciate for the advice. We highlighted the amendments instead of using track-changes mode.

---

## [Editor Report · Decision Letter 4]

6 Jan 2021

DNA-thioguanine nucleotide as a treatment marker in acute lymphoblastic leukemia patients with *NUDT15* variant genotypes

PONE-D-20-27853R4

Dear Dr. Koo,

We’re pleased to inform you that your manuscript has been judged scientifically suitable for publication and will be formally accepted for publication once it meets all outstanding technical requirements.

Kind regards,

A. M. Abd El-Aty

Academic Editor

PLOS ONE
---

## [Editor Report · Acceptance letter]

12 Jan 2021

PONE-D-20-27853R4 

DNA-thioguanine nucleotide as a treatment marker in acute lymphoblastic leukemia patients with *NUDT15* variant genotypes 

Dear Dr. Koo:

I'm pleased to inform you that your manuscript has been deemed suitable for publication in PLOS ONE. Congratulations! Your manuscript is now with our production department. 

Kind regards, 

on behalf of

Prof. A. M. Abd El-Aty 

Academic Editor

PLOS ONE